# Spike4DGS: Towards High-Speed Dynamic Scene Recontruction with 4D Gaussian Splatting via a Spike Camera Array

**Qinghong Ye**[1,3,*], **Yiqian Chang**[2,3,*], **Jianing Li**[4], **Haoran Xu**[3,5], **Xuan Wang**[2] ,
**Wei Zhang**[3,†], **Yonghong Tian**[1,3,4,†], **Peixi Peng**[1,3,†]

[1]Shenzhen Graduate School of Peking University
[2] Shenzhen Campus of Harbin Institute of Technology
[3]Peng Cheng Laboratory
[4] School of Computer Science, Peking University
[5] Shenzhen Campus of Sun Yat-sen University

## Abstract

Spike camera with high temporal resolution offers a new perspective on high-speed dynamic scene rendering. Most existing rendering methods rely on Neural Radiance Fields (NeRF) or 3D Gaussian Splatting (3DGS) for static scenes using a monocular spike camera. However, these methods struggle with dynamic motion, while a single camera suffers from limited spatial coverage, making it challenging to reconstruct fine details in high-speed scenes. To address these problems, we propose Spike4DGS, the first high-speed dynamic scene rendering framework with 4D Gaussian Splatting using spike camera arrays. Technically, we first build a multi-view spike camera array to validate our solution, then establish both synthetic and real-world multi-view spike-based reconstruction datasets. Then, we design a multi-view spike-based dense initialization module that obtains dense point clouds and camera poses from continuous spike streams. Finally, we propose a spike-pixel synergy constraint supervision to optimize Spike4DGS, incorporating both rendered image quality loss and dynamic spatiotemporal spike loss. The results show that our Spike4DGS outperforms state-of-the-art methods in terms of novel view rendering quality on both synthetic and real-world datasets. More details are available at the **project page**.

## 1 Introduction

Novel view synthesis [16] is a cornerstone of many cutting-edge applications, enabling the creation of precise novel views from ideal image sequences. However, conventional cameras struggle in high-speed motion scenarios, where rapid movement causes motion blur and significantly degrades reconstruction quality [39]. While some approaches [30, 44] attempt to improve reconstruction from motion-blurred images, they remain fundamentally limited by the sampling rates of RGB cameras. As a result, the use of new vision sensors for high-quality rendering in high-speed scenes has garnered increasing attention.

Spike cameras [9, 20, 48] with high temporal resolution offer a new perspective on high-speed scene rendering. Unlike conventional cameras, they asynchronously encode absolute light intensity into continuous spike streams at rates of up to 20k Hz. This unique property makes them particularly

---

*Qinghong Ye and Yiqian Chang contributed equally to the paper.

†Peixi Peng,Yonghong Tian({pxpeng,yhtian}@pku.edu.cn ), and Wei Zhang(zhangwei1213052@126.com) are corresponding authors.

effective for preserving high-speed scene textures with greater detail. Existing studies [64, 65, 24, 67, 66, 62, 60, 58, 57, 46, 59, 47] have demonstrated their capability for fine-grained 2D reconstruction. The advantages of spike cameras also indicate their immense potential for advancing 3D scene reconstruction and novel view synthesis.

Extensive works [15, 61] have explored neuromorphic cameras for rendering in high-speed scenarios. Some studies [19, 38, 21, 28, 3, 31, 52, 36] utilize Radiance Fields (NeRF) and 3D Gaussian Splatting (3DGS) for scene representation and novel view synthesis using event cameras. However, event cameras capture only light changes rather than absolute brightness, making them challenging for fine-grained 3D reconstruction. Other efforts [69, 27, 18, 51, 54] have explored 3DGS and NeRF with alternative spike cameras to overcome these limitations and enhance rendering quality. For example, pioneering works with SpikeNeRF [69] and Spike-NeRF [18] have demonstrated the feasibility of using spike streams to reconstruct 3D scenes. Nevertheless, NeRF-based methods suffer from time-consuming training and inference processes, and their implicit representations limit scene editing capabilities. In contrast, Gaussian Splatting offers a compelling alternative, providing high accuracy and fast inference speeds. For instance, SpikeGS [51, 54], SpikeNVS [7], and USP-Gaussian [5] achieve impressive results in photorealistic 3D reconstruction using spike cameras. However, these 3DGS-based methods may face difficulties in handling high-speed dynamic scenes. Additionally, relying solely on monocular cameras may present 3D reconstruction challenges in areas with weak textures or when using static cameras. In fact, camera arrays could enhance texture information through multi-view perspectives, enabling dynamic scene rendering even without camera motion.

To address the aforementioned challenges, we propose Spike4DGS, the first high-speed dynamic scene rendering framework utilizing 4D Gaussian Splatting with spike camera arrays. We aim at overcoming the following challenges: (i) *Camera setup and dataset* – How could we establish a multi-view spike camera array and build high-quality spike-based reconstruction datasets? (ii) *Effective model* – How could we design an efficient dynamic scene rendering model that directly processes multi-view spike streams with 4D Gaussian splatting(4DGS)?

To be specific, we first build a multi-view spike camera array and then establish both synthetic and real-world spike-based reconstruction datasets. Then, we design a novel multi-view spike-based dense initialization module to generate dense point clouds and camera poses from continuous spike streams. Finally, we propose a pixel-spike synergy supervision strategy to optimize Spike4DGS, which incorporates both reconstructed image quality loss and dynamic spatiotemporal spike loss. Experimental results show that our Spike4DGS outperforms state-of-the-art methods in terms of rendering quality on both synthetic and real-world datasets. We further verify that spike cameras achieve higher rendering quality than event cameras and RGB cameras in high-speed scenes. Meanwhile, rendering quality improves as the number of spike cameras increases.

The main contributions of this work are summarized as:

- We introduce Spike4DGS, the first framework that combines spike camera arrays with 4DGS, enabling novel view synthesis in high-speed dynamic scenarios.

- We present a Spike-Pixel Synergy Supervision strategy to optimize the parameters of our Spike4DGS for enhanced rendering quality.

- We build a spike camera array, along with a highly realistic synthetic and real-world datasets that contain multi-view spike streams. We believe two standardized datasets open up opportunities for research in this novel problem.

## 2 Related Work

### 2.1 NVS for Dynamic Scenes

Novel View Synthesis (NVS) tasks aim to generate unknown views of an object or scene from a set of images of known views. Representative papers include Neural Radiance Fields (NeRF) [32] and 3D Gaussian Splatting (3DGS) [23]. Recently, a large number of static 3DGS-based techniques [4, 22, 53] have been proposed due to their high quality and real-time rendering without using neural networks like NeRF. However, the assumption of static scenes prevents application to real-world scenarios with moving objects. Therefore, several works extend the 3DGS to dynamic scenes[45, 50, 29, 26, 2]. These methods are usually divided into two main lines. For instance, De3DGS [50] and 4DGS [45]

model spatial-temporal deformation with an implicit deformation field as the first line. On the other hand, the second line is based on the idea that scenes' motion could be encoded into the 3D Gaussian representation straightly, such as STG [26] and D3DGS [29], which represents changes in 3D Gaussian over time through a temporal opacity and a polynomial function for each Gaussian. The above approaches could perform well on synthetic datasets and simple real-world datasets. However, when there are some high-speed objects in the scene, traditional dynamic reconstruction methods using RGB image data may suffer motion blur since most standard RGB cameras have limited frame rates. This motivates us to use the neuromorphic cameras to avoid this problem.

## 2.2 Neuromorphic Cameras on 3DGS

There are two types of bio-inspired sensors: event cameras [15] and spike cameras [9]. Event cameras are based on the temporal contrast sampling method and generate events asynchronously when pixel brightness changes exceed a threshold. Early event-based reconstruction approaches [28, 3, 31, 38, 21, 42] have been proposed to derive neural radiance fields directly from event streams. E2nerf [36] and evagaussian [52] achieved sharp reconstruction from blurry images. E-4DGS [13] achieved high-fidelity dynamic reconstruction from the multi-view event cameras. GS2E [25] has introduced an effective event stream generator by gaussian splatting. Another type of neuromorphic camera is called the spike camera. Spike cameras record the absolute light intensity at a fairly high frame rate and provide a more explicit input format for detailed reconstruction. Some nerf-based methods [69, 27, 18] have verified the feasibility of reconstruction with spike, but they suffer suboptimal training and rendering speeds due to the complex spike simulation network. Some 3DGS-based spike reconstruction methods have emerged to optimize this defect. Yu's SpikeGS[51] reconstructed view synthesis results from a continuous spike stream captured by a moving spike camera. In a harder setting, Zhang's SpikeGS[54] reconstructed scenes via a single spike stream with monocular high-speed camera motion. However, there is no established spike-based method for addressing the challenge of rendering high-speed dynamic scenes using multi-view spike streams. On this basis, our work aims to overcome the limitations and construct a spike-based 3D Gaussian Splatting model for high-speed dynamic scenes via a spike camera array.

# 3 Methodology

## 3.1 Preliminaries

**Spike Camera.** Spike camera is a bio-inspired sensor which records and converts the absolute light intensity at a fairly high frame rate (up to 20kHz) into accumulated voltage through photoreceptors [64, 8]. If the accumulated voltage $V$ reaches the scheduling threshold $\Theta$, a spike will be triggered and $V$ is reset to zero, mathematically formulated as follows:

$$V(t) = \int_{t_s}^{t} \sigma \cdot L(t) dt \, \mathrm{mod} \Theta, \tag{1}$$

where $L(t)$ represents the instant light intensity at time $t$, $t_s$ is the moment when the previous spike was emitted, and $\sigma$ is the constant photoelectric conversion coefficient.

**DUSt3R Initialization.** DUSt3R [41] is a dense initialization method used for 3D reconstruction. Compared with COLMAP Initialization [40], DUSt3R provides more accurate point clouds under low-quality image input with less time, which is more suitable for high-speed scenes. Specifically, given a pair of images $(I_1, I_2)$, DUSt3R utilizes a ViT for the encoder and decoder [11] and a DPT head [37] for estimating point clouds $\mathcal{PC}$:

$$\mathcal{PC} = \mathrm{DPT}(\mathrm{ViT}(I_1, I_2)). \tag{2}$$

Although dUST3r and its improved versions [49, 55] could generate dense point clouds based on multi-view images, however, the quality of point clouds depends on the quality of input images.

**4D Gaussian Splatting.** 4D Gaussian Splatting (4DGS) [45] is used for rendering dynamic scenes. It proposes a network that learns the Gaussian deformation field to predict the deformation of each 3D Gaussian. For input 3D Gaussian $\mathcal{G}$ and time $t$, a spatial-temporal structure encoder $\mathcal{H}$ and a multi-head Gaussian deformation decoder $\mathcal{D}$ are used for calculating the deformations $\Delta\mathcal{G}$:

$$\Delta\mathcal{G} = \mathcal{D}(\mathcal{H}(\mathcal{G}, t)). \tag{3}$$

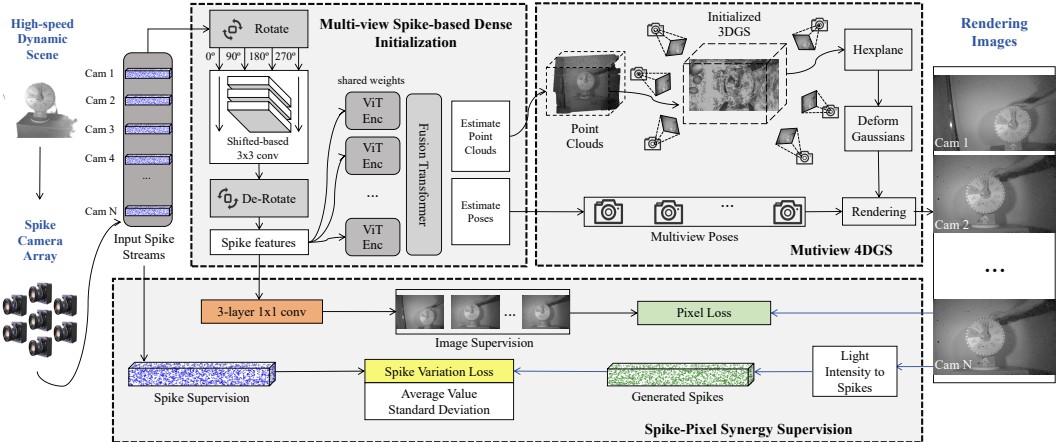

Figure 1: The Framework of our Spike4DGS, we establish the connection between the real-world spike streams and the dynamic scene rendering images. The input multi-view spike streams are sent to a **Multi-view Spike-based Dense Initialization** to estimate point cloud and camera poses. Based on the initialization, a 4DGS with **Spike-Pixel Synergy Supervision** consisting of a Pixel Loss and a Spike Variation Loss is utilized for rendering. Our Spike4DGS could reconstruct high-speed dynamic scenes with delicate motion and texture details.

## 3.2 Problem Fomulation

We aim to reconstruct high-speed dynamic scenes with delicate motion and texture details. To reach this goal, we build a multi-view spike camera array to capture the high-speed motion and propose a novel method called Spike4DGS for rendering based on continuous spike streams. The problem could be denoted as:

$$\textbf{Spike4DGS} : \{(S_1, S_2, ..., S_{N-1}), V_N, t\} \rightarrow \hat{I}_t \tag{4}$$

where $S_1, S_2, ..., S_{N-1}$ are the spike streams captured from N-1 views of spike cameras for training, $\hat{I}_t$ is the rendering image of novel view $V_N$ (the view of $N$-th spike camera) at time $t$.

To solve this problem, Spike4DGS first builds an end-to-end Multi-view Spike-based Dense Initialization (MSDI) method to estimate the dense point cloud and camera poses from input spike streams, as detailed in Sec. 3.3. Then, the initial point cloud and camera poses are sent to a 4D Gaussian Splatting to generate novel view rendering results. To get more delicate motion and texture details, Spike-Pixel Synergy Supervision (SPSS) is proposed for the constraint of 4D Gaussian Splatting in Sec. 3.4. The total framework of Spike4DGS is shown in Fig. 1.

## 3.3 Multi-view Spike-based Dense Initialization

Previous rendering tasks based on spike cameras, such as SpikeNeRF [69], employ a two-step initialization method. They first use spike-to-image methods such as TFI [64], TFP [64] and spk2img [6] to get images, and then utilize COLMAP [40] or DUSt3R [41] on these images for the scene initialization. However, this two-step method is complex and requires high-quality spike-to-image results. When dealing with high-speed dynamic scenes, the lack of texture details in converted images may influence the final point cloud estimation. In contrast, we propose an end-to-end framework, called Multi-view Spike-based Dense Initialization (MSDI), which consists of a spike feature extractor, a point cloud and pose estimator, and an image generator. Given a series of spike streams from $N$ spike cameras, MSDI aims to estimate 3D point clouds, camera poses, and their corresponding images:

$$\textbf{MSDI} : (S_1, ..., S_N) \rightarrow \{\mathcal{PC}, ([R|T]_1, ..., [R|T]_N), (I_1, ..., I_N)\}. \tag{5}$$

where $\mathcal{PC}$ is the estimated point cloud, $[R|T]_1, ..., [R|T]_N$ are the camera rotation and translation parameters of $N$ spike cameras, $I_1, ..., I_N$ are the images generated from input spike streams. As

an end-to-end network, MSDI is fine-tuned together with pre-trained weights on multi-view spike streams captured from the Carla [10] simulator.

**Spike Feature Extractor.**    Firstly, we build a feature extractor for the multi-view spike streams. Assuming $\Gamma_t$ is a time interval around frame time $t$. For an input spike stream $S_i(\Gamma)$ which lasts for a time interval $\Gamma_t$ and is captured from the $i$-th camera view, MSDI rotates $S_{\Gamma_t}$ four times to obtain a complete receptive field in four directions and concentrate them together:

$$R(S_i(\Gamma_t)) = \text{CAT}\{\text{Rot}(S_i(\Gamma_t), \theta)\}|\theta \in \{0°, 90°, 180°, 270°\}, \tag{6}$$

where $\text{CAT}$ and $\text{Rot}$ mean concatenation and rotation operations respectively. Then, we utilize a shift-based $3 \times 3$ convolution layer from [54] to extract features:

$$\hat{f}_i(\Gamma_t) = \mathcal{M}(R(S_i(\Gamma_t))), \tag{7}$$

where $\mathcal{M}$ is the shift-based convolution layer, $\hat{f}_{\Gamma_t}$ is the extracted spike features for input spike streams $S_i(\Gamma_t)$. Replacing the input $S_i(\Gamma_t)$ to $N$ views of spike streams, we can get a series of extracted features $(\hat{f}_1(\Gamma_t), ..., \hat{f}_N(\Gamma_t))$.

**Point Cloud and Pose Estimator.**    To estimate the point cloud and camera poses, the problem could be formulated as:

$$\text{Estimator} : (\hat{f}_1(\Gamma_t), ..., \hat{f}_N(\Gamma_t)) \to \mathcal{PC}, ([R|T]_1, ..., [R|T]_N). \tag{8}$$

To solve the problem, MSDI builds an estimator consisting of $N$ ViT encoders [11] (equals to the number of views) and a fusion transformer. For $N$ ViT encoders, each encoder $\text{ViT}_i$ handles a camera view with shared weights:

$$\hat{F}_i(\Gamma_t) = \text{ViT}_i(\hat{f}_i(\Gamma_t)), i \in (1, ..., N). \tag{9}$$

For the fusion transformer $\text{FusionTF}$, it is a 24-layer transformer which is the same as [49]:

$$G_1(\Gamma_t), G_2(\Gamma_t), ..., G_N(\Gamma_t) = \text{FusionTF}\left(\hat{F}_1(\Gamma_t), \hat{F}_2(\Gamma_t), ..., \hat{F}_N(\Gamma_t)\right), \tag{10}$$

where $G_i(\Gamma_t)$ is the temporal feature of $i$-th camera view. This operation generates temporal features with global contextual understanding from all views.

Then, a DPT [37] decoding head is utilized for decoding the temporal features into a point cloud $\mathcal{PC}$ and its corresponding confidence map $\Sigma_{\mathcal{PC}}$:

$$(\mathcal{PC}, \Sigma_{\mathcal{PC}}) = \text{DPT}(G_1(\Gamma_t), G_2(\Gamma_t), ..., G_N(\Gamma_t)). \tag{11}$$

Finally, we utilize the global point cloud $\mathcal{PC}$ to estimate camera rotations and translations $([R|T]_1, ..., [R|T]_N)$ via RANSAC-PnP [14, 1].

**Image Generator.**    In addition, MSDI also generates discrete images from continuous spike streams. For the time interval $\Gamma_t$ around frame time $t$, the image at time $t$ in $i$-th camera view could be generated from:

$$\bar{I}_i(t) = \text{BCONV}(\hat{f}_i(\Gamma_t)), i \in (1, ..., N), \tag{12}$$

where $\bar{I}_i(t)$ is the generated image, $\text{BCONV}$ is a network consisting of three $1 \times 1$ convolutions followed from BSN [6].

Compared with the previous two-step initialization methods like TFI [64]+COLMAP [40], our end-to-end MSDI method could avoid the errors of the final estimations which occur from the low-quality images generated in the intermediate steps.

## 3.4   4DGS with Spike-Pixel Synergy Supervision

After MSDI, we could get the initial point cloud and camera poses. Then the initial 3D Gaussian $\mathcal{G}$ could be obtained from them. Thus, we could generate the deformed Gaussians from 4DGS [45] and render an image $\hat{I}(t)$ at $i$-th view $p_i$:

$$\hat{I}_i(t) = \text{Render}(4\text{DGS}(\mathcal{G}, t), p_i). \tag{13}$$

According to 4DGS, the rendered image loss could be formulated as follows to offer our Spike4DGS pixel supervision:

$$\mathcal{L}_t^{\text{pixel}} = ||\hat{I}_i(t) - \bar{I}_i(t)||_1, \tag{14}$$

where $\hat{I}_i(t)$ is the rendered image at the time $t$, $\bar{I}_i(t)$ is the generated images from the above MSDI. However, this pixel loss concentrates only on image similarity but ignores the texture and motion details, which are contained in spike streams. To take advantage of the spike characteristics, we propose a spike variation loss which first translates the rendered images into spike streams and then compares the variation between generated and real spike streams.

**Translate from Rendered Image to Spike Stream.** Let us denote the intensity values of the pixel $(x, y)$ in rendered images at time $t$ as $\hat{I}_i(x, y, t)$. After getting the real light of the scene, we convert the scene light intensity into spike streams using an Integrate-and-Fire (IF) [17, 12] mechanism. Following intensity translation method [68], we could establish the following relationship:

$$\hat{S}_i(x, y, t) = \text{IF}(\hat{I}_i(x, y, t) \cdot n(x, y)), \tag{15}$$

where $n(x, y)$ is the deviation matrix corresponding to the response nonuniformity noise which could be obtained by capturing a uniform light scene and recording the intensity. $\text{IF}(\cdot)$ is a IF neuron, and $\hat{S}_i(x, y, t)$ is the predicted spike values of the pixel $(x, y)$ at time $t$.

**Spike Variation Supervision.** Different from static scenes, objects in high-speed moving scenes are constantly changing, thus the segment of spikes $S_i(\Gamma_t)$ in a time interval $\Gamma_t$ around $t$ is a continuously changing sequence. For high-speed moving objects, naive L1 or L2 loss treats each frame separately, without considering how these objects move over time. In contrast, we propose spike variation supervision, in order to concentrate on the dynamic changes of objects. Since high-speed motions are continuous, spike streams in a short time interval around time $t$ could be generated from a single spike at time $t$. Thus, we simply design a one-layer MLP network $\phi_s$ to map a single spike $\hat{S}_i(t)$ to a spike sequence $\hat{S}_i(\Gamma_t)$ in a time interval $\Gamma_t$ around $t$:

$$\hat{S}_i(\Gamma_t) = \phi_s(\hat{S}_i(t)), \tag{16}$$

where the shape of $\hat{S}_i(t)$ is $(H, W, 1)$ and the shape of $\hat{S}_i(\Gamma_t)$ is $(H, W, \Gamma_t)$. $H$ and $W$ are the height and width of the spikes. Then, we calculate the average value and the standard deviation of this spike sequence $\hat{S}_i(\Gamma_t)$ and compare them with ground truth. Therefore, the Spike Variation Loss could be presented as:

$$\mathcal{L}_t^{\text{spikeV}} = ||\text{avg}(\hat{S}_i(\Gamma_t)) - \text{avg}(S_i^{\text{gt}}(\Gamma_t))||_1 + ||\text{std}(\hat{S}_i(\Gamma_t)) - \text{std}(S_i^{\text{gt}}(\Gamma_t))||_1. \tag{17}$$

Combining this regularization with the original Pixel Loss, we get the final synergy training loss for our Spike4DGS:

$$\mathcal{L}_t^{\text{total}} = \mathcal{L}_t^{\text{spikeV}} + \mathcal{L}_t^{\text{pixel}}. \tag{18}$$

## 4 Experiment

### 4.1 Datasets

To verify the validity of our proposed Spike4DGS, we create two datasets. The first dataset is a real-world object dataset collected by the aforementioned spike camera array. The second dataset is a high-speed synthetic outdoor dataset generated by the CARLA simulator [10]. All our experiments are conducted on these two self-made datasets. We achieve the best results on both two datasets, which demonstrates the superiority of our Spike4DGS.

**Real-world Object Dataset.** As shown in Fig. 2, the spike array consists of 9 spike cameras that are capable of capturing spike streams with a spatial resolution of $250 \times 400$ and a temporal resolution of 20k Hz. During the data collection process, synchronized recordings were made from all 9 cameras, ensuring that motion was consistently represented across different

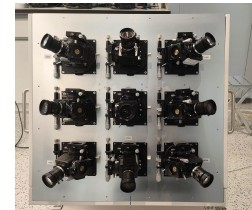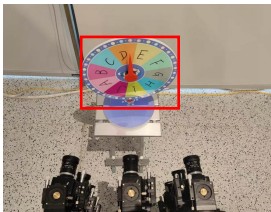

Figure 2: Our spike camera array.

Table 1: Quantitative evaluation on synthetic outdoor dataset.Unit: PSNR-dB $\uparrow$, SSIM $\uparrow$, LPIPS $\downarrow$.

| Method | Jaywalk | | | Bicycle | | | Motor | | | Car | | | Van | | |
|---|---|---|---|---|---|---|---|---|---|---|---|---|---|---|---|
| | PSNR | SSIM | LPIPS | PSNR | SSIM | LPIPS | PSNR | SSIM | LPIPS | PSNR | SSIM | LPIPS | PSNR | SSIM | LPIPS |
| TFI[64]+D3DGS[29] | 20.65 | 75.7 | 0.384 | 21.73 | 77.7 | 0.385 | 19.98 | 76.2 | 0.377 | 20.09 | 76.2 | 0.387 | 19.24 | 75.3 | 0.402 |
| TFP[67]+D3DGS[29] | 20.88 | 76.5 | 0.357 | 21.78 | 78.9 | 0.346 | 19.85 | 76.0 | 0.372 | 20.07 | 76.6 | 0.361 | 19.20 | 76.4 | 0.389 |
| Spk2img[57]+D3DGS[29] | 20.94 | 76.6 | 0.304 | 21.68 | 77.5 | 0.379 | 20.12 | 76.5 | 0.380 | 20.15 | 76.3 | 0.389 | 18.93 | 75.9 | 0.392 |
| TFI[64]+STG[26] | 24.48 | 84.2 | 0.224 | 24.52 | 84.1 | 0.221 | 24.47 | 84.3 | 0.227 | 23.50 | 84.0 | 0.223 | 26.55 | 88.7 | 0.201 |
| TFP [67]+STG[26] | 24.45 | 84.0 | 0.220 | 24.50 | 84.2 | 0.225 | 24.53 | 84.1 | 0.222 | 24.60 | 84.3 | 0.226 | 26.48 | 89.0 | 0.205 |
| Spk2img[57]+STG[26] | 24.72 | 84.6 | 0.221 | 24.75 | 84.5 | 0.220 | 24.70 | 84.7 | 0.223 | 23.73 | 84.8 | 0.219 | 25.94 | 88.4 | 0.202 |
| TFI[64]+4DGS[45] | 26.88 | 87.7 | 0.213 | 27.02 | 89.7 | 0.202 | 25.52 | 90.5 | 0.198 | 25.38 | 86.6 | 0.219 | 25.21 | 86.1 | 0.214 |
| TFP [67]+4DGS[45] | 27.96 | 91.2 | 0.192 | 27.15 | 90.4 | 0.196 | 26.57 | 88.3 | 0.206 | 25.01 | 86.5 | 0.213 | 25.27 | 87.0 | 0.220 |
| Spk2img[57]+4DGS[45] | 27.04 | 90.9 | 0.208 | 26.17 | 88.6 | 0.219 | 26.35 | 88.9 | 0.218 | 24.75 | 87.8 | 0.229 | 24.87 | 87.9 | 0.227 |
| Dy-SpikeGS[51] | 23.04 | 0.852 | 0.243 | 22.17 | 0.846 | 0.249 | 23.35 | 0.857 | 0.238 | 22.75 | 0.849 | 0.244 | 22.87 | 0.851 | 0.242 |
| Dy-SpikeNeRF[69] | 17.04 | 0.784 | 0.372 | 16.17 | 0.773 | 0.384 | 16.35 | 0.776 | 0.379 | 14.75 | 0.759 | 0.395 | 15.87 | 0.768 | 0.388 |
| **Our Spike4DGS** | **28.29** | **93.1** | **0.189** | **27.69** | **91.3** | **0.185** | **27.92** | **91.6** | **0.193** | **27.74** | **91.2** | **0.199** | **27.13** | **90.1** | **0.197** |

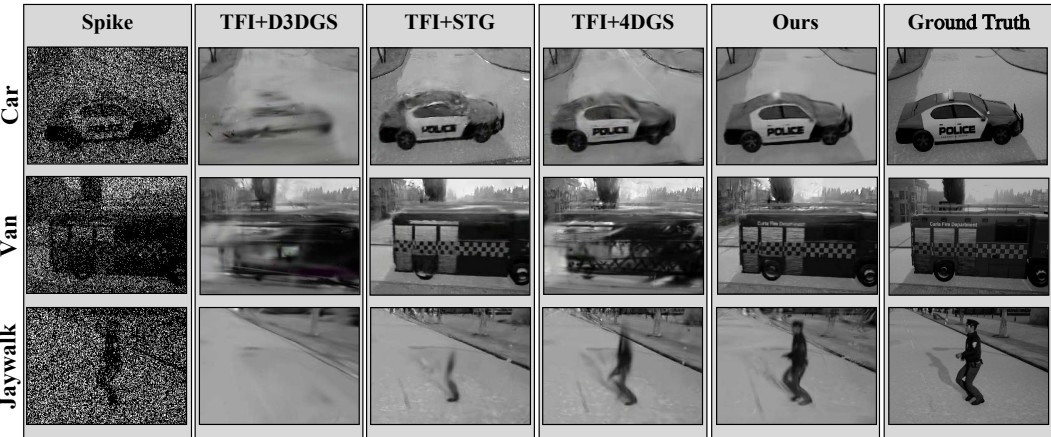

Figure 3: Quantitative comparison with other methods on the dataset on our synthetic outdoor dataset. We mainly compare our method with some SOTA approaches. In contrast, our method delivers both superior outlines and clear details.

views. When capturing spike streams, we fix the position of our spike camera array and place the high-speed dynamic objects in front of the cameras.Then we record 9 spike streams of approximately 0.5 seconds for each real-world scene with high-speed objects. Firstly, we choose high-speed dynamic objects such as "the collapse of bricks" (Bricks) and "the spin of a turntable"(Turntable) and put them before our spike camera array.Secondly, we minimize noise by providing the spike camera with ideal light intensity and obtaining multiple ideal spike streams. These spike streams could be converted to images by our MSDI module in the training process, and we use them as image supervision.

**Synthetic Outdoor Dataset.** To quantitatively analyze our superiority, we create a synthetic outdoor dataset using the Carla [10] simulator, which includes scenarios like Jaywalk, Bicycle, Van, Car, and Motor. These scenarios feature objects of varying sizes and speeds, with objects appearing from the left side of the image and moving to the right. An array of 9 camera sets with an overhead view was set up to capture different views of the objects. Each view setting consists of a spike camera at 20k FPS for training and an RGB camera at 1k FPS for evaluation.

## 4.2 Experimental Setup

**Competitors.** Due to the relative lack of methods for dynamic novel view synthesis based on spike cameras, we completed the comparison using some two-stage rendering approaches. First, we choose some direct spike-to-image approaches: TFI[64], TFP[64], and spk2img[57], and then combine them with previous multiview dynamic NVS methods[45, 26, 29] .For a more comprehensive experiment, we also manually integrated a deformation network from 4DGS into SpikeGS and SpikeNeRF (denoted as Dy-SpikeGS and Dy-SpikeNeRF) as comparison. Those comparisons are initialized by

Table 2: Quantitative evaluation on real-world object dataset. Unit: Brisque ↓, NIQE ↓, MetaIQA ↑.

| Method | Bricks | | | Chips | | | Turntable | | | Bird | | |
|---|---|---|---|---|---|---|---|---|---|---|---|---|
| | Brisque | NIQE | MetaIQA | Brisque | NIQE | MetaIQA | Brisque | NIQE | MetaIQA | Brisque | NIQE | MetaIQA |
| TFI[64]+D3DGS[29] | 57.35 | 16.71 | 0.114 | 61.32 | 16.43 | 0.124 | 56.53 | 15.34 | 0.117 | 57.87 | 14.87 | 0.112 |
| TFP [67]+D3DGS[29] | 56.87 | 15.37 | 0.127 | 60.15 | 15.01 | 0.131 | 55.57 | 14.93 | 0.123 | 56.37 | 14.53 | 0.126 |
| Spk2img[57]+D3DGS[29] | 58.78 | 15.98 | 0.133 | 59.27 | 15.04 | 0.135 | 56.94 | 15.87 | 0.129 | 57.14 | 14.65 | 0.137 |
| TFI[64]+STG [26] | 45.33 | 13.88 | 0.143 | 43.77 | 13.93 | 0.149 | 36.45 | 10.53 | 0.150 | 33.32 | 12.64 | 0.143 |
| TFP [67]+STG [26] | 44.83 | 13.21 | 0.148 | 44.46 | 11.08 | 0.150 | 37.94 | 10.87 | 0.161 | 34.09 | 9.98 | 0.155 |
| Spk2img[57]+STG [26] | 45.61 | 13.63 | 0.147 | 44.67 | 12.08 | 0.152 | 38.01 | 10.92 | 0.157 | 34.02 | 9.27 | 0.149 |
| TFI[64]+4DGS [45] | 39.56 | 10.23 | 0.165 | 45.15 | 10.01 | 0.162 | 37.57 | 9.93 | 0.170 | 33.37 | 9.53 | 0.163 |
| TFP [67]+4DGS [45] | 40.58 | 10.42 | 0.164 | 44.46 | 11.08 | 0.150 | 37.94 | 10.87 | 0.161 | 34.09 | 9.98 | 0.155 |
| Spk2img[57]+4DGS [45] | 40.53 | 10.55 | 0.167 | 44.67 | 12.08 | 0.152 | 38.01 | 10.92 | 0.157 | 34.02 | 9.27 | 0.149 |
| Dy-SpikeGS[51] | 48.73 | 14.23 | 0.136 | 49.15 | 13.84 | 0.139 | 47.92 | 13.57 | 0.133 | 46.88 | 13.22 | 0.137 |
| Dy-SpikeNeRF[69] | 61.84 | 17.25 | 0.116 | 62.17 | 16.93 | 0.113 | 60.58 | 16.71 | 0.118 | 59.74 | 16.35 | 0.115 |
| **Our Spike4DGS** | **34.29** | **9.95** | **0.176** | **33.52** | **8.03** | **0.183** | **26.86** | **9.86** | **0.179** | **23.86** | **8.36** | **0.180** |

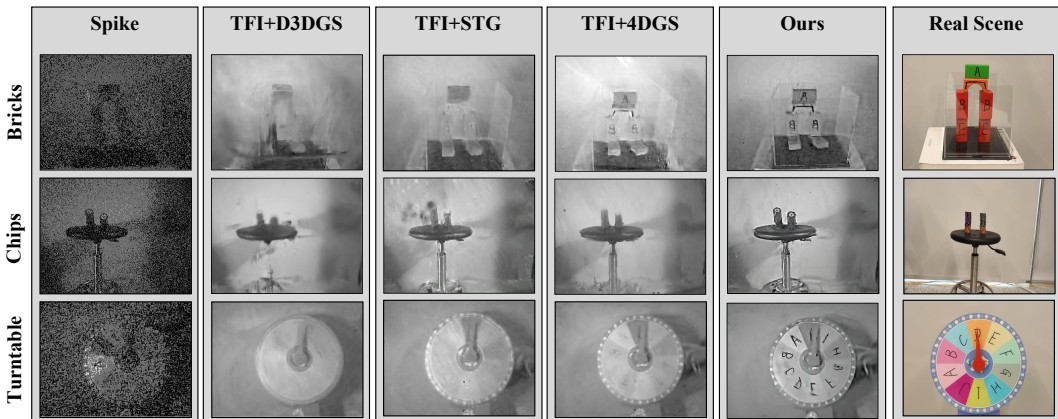

Figure 4: Qualitative comparison on real-world object datasets with SOTA rendering methods. We present the RGB photos of real scenes in the last column due to the lack of ground truth.

DUSt3R [41] to obtain point clouds. In our framework, the training data are multiple spike streams for both synthetic and real datasets.

**Implementation Details.** Firstly, we use the MSDI in Sec. 3.3 to estimate initialize point clouds, camera poses, and generate images. We fine-tune this end-to-end framework together with pre-trained weights using multi-view spike streams captured from dynamic scenes in the Carla simulator. The fine-tuning adopts L1 loss supervised by Carla images and confidence-aware pointmap regression loss in DUSt3R [41]. We utilize Adam optimizer with a learning rate of 0.0001 to effectively minimize both losses and ensure stable convergence during training. Secondly, for 4DGS with Spike-Pixel Synergy supervision in Sec. 3.4, the learning rate is the same as 4DGS [45]. At each optimization iteration, we randomly sample a batch of views from the same time t. The total experiments are conducted on a single NVIDIA GTX 4090 with PyTorch and the optimization for a single scene typically takes about 20 minutes to converge and 40 FPS when rendering. For the metrics of the synthetic outdoor dataset, we employ three widely-used image quality assessment metrics, PSNR [45], SSIM [43] and LPIPS [56]. For the real dataset, which lacks corresponding ground truth images, we employ NIQE [35], BRISQUE [33, 34] and MetaIQA [63] as no-reference image quality evaluation metrics, the same as state-of-the-art works [69, 51].

### 4.3 Performance Comparison

#### 4.3.1 Synthetic Outdoor Data Experiments

**Quantitative Performance.** We present a detailed comparison of our method against some two-stage rendering approaches, such as TFI [64]+STG [26], TFI[64]+D3DGS[29], TFI[64]+4DGS [45], on our synthetic outdoor dataset. As demonstrated in Table 1, our method outperforms the SOTA two-stage rendering methods across all five distinct scenes. Note that our Spike4DGS improves more on the higher-speed scenes like "Car", which proves the effect in high-speed scenarios.

**Qualitative Performance.** We also present the qualitative results in Fig. 3, where: (I) TFI [64]+D3DGS [29] almost failed to reconstruct the outlines and details of each scene. (II) TFI [64]+4DGS [45] could only reconstruct the outlines with texture details missing. (iii) In contrast, our method demonstrates superior performance, generating clearer and more detailed novel views, especially in challenging high-speed scenes like "Van".

### 4.3.2 Real-world Object Data Experiments

**Quantitative Performance.** As shown in Table 2, our method achieves superior performance compared to those SOTA rendering approaches. Our method achieves an average improvement in BRISQUE [33, 34], NIQE [35] and MetaIQA [63] **by 15.4%, 4.6% and 6.7%** respectively, indicating higher-quality novel view rendering.

**Qualitative Performance.** In qualitative experiments, we focus on the generated texture details. Fig. 4 presents that our method could generate high-quality texture details. For example, in the third line of Fig. 4, our rendering could **generate clear alphabets "A, B, C. . ."** on the Turntable dataset while others could not.

### 4.3.3 Analysis

**Performance Analysis.** The quantitative and qualitative results above show that our Spike4DGS could significantly surpass the SOTA approaches. Moreover, on the synthetic outdoor dataset, our method proves the ability to render the outdoor high-speed scenes. While on the real-world object dataset, our method could also reconstruct indoor high-speed objects. This highlights the robustness and comprehensive improvements of our Spike4DGS on different high-speed dynamic scenes.

### 4.4 Ablation Study

**Contribution of Each Component.** We conduct an ablation study to assess the contribution of each component in our Spike4DGS. As shown in Table 3, we combine 4DGS with some initialization frameworks. The results demonstrate that our MSDI initialization achieves the best performance, and the SPSS module consistently improves reconstruction quality under different initialization frameworks. Results in the last row demonstrate that the full model (4DGS+MSDI+SPSS) achieves the best performance, validating the effectiveness of our design for spike-based 3D reconstruction.

**Ablation on Supervision.** To evaluate the effect of our supervision strategy, we compare our SPSS in Sec. 3.4 with pixel loss in vanilla 4DGS [45], spike loss in SpikeNerf [69], and the combination of the above. The quantitative results of novel view synthesis are listed in Table 4. Our SPSS, consisting of pixel loss and spike variation loss, obtains the highest PSNR performance.

**Ablation on View Numbers.** In this part, we investigate the relationship between view numbers and performance. As shown in Fig. 5, we present both quantitative and qualitative comparisons. For the quantitative experiments, we use Brisque [33, 34] and NIQE [35] as the evaluation metric. Our Spike4DGS achieves the best performance

Table 3: The contribution of each component.

| Methods | PSNR↑ | SSIM↑ | LPIPS↓ |
|---|---|---|---|
| TFI [64]+COLMAP [40] | 21.08 | 84.9 | 0.253 |
| TFI [64]+DUSt3R [41] | 25.66 | 89.3 | 0.205 |
| MSDI | 26.66 | 90.3 | 0.198 |
| TFI [64]+COLMAP [40]+SPSS | 22.76 | 85.2 | 0.233 |
| TFI [64]+DUSt3R [41]+SPSS | 26.34 | 90.2 | 0.201 |
| **MSDI+SPSS(Full model)** | **27.74** | **91.2** | **0.190** |

Table 4: Ablation on supervision strategy.

| Supervision Strategy | PSNR↑ | SSIM↑ | LPIPS↓ |
|---|---|---|---|
| PixelLoss [45] | 21.22 | 85.0 | 0.251 |
| SpikeLoss [69] | 13.79 | 76.3 | 0.408 |
| PixelLoss [45]+SpikeLoss [69] | 26.69 | 89.5 | 0.203 |
| **SPSS (Ours)** | **27.74** | **91.2** | **0.190** |

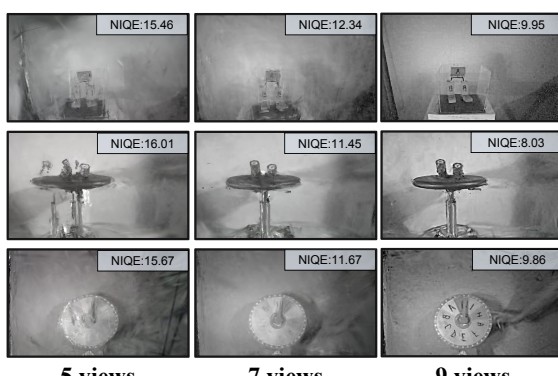

**5 views**  **7 views**  **9 views**

Figure 5: Quantitative and qualitative ablation of view numbers on real-world object datasets.

Table 5: Comparison of different methods under varying numbers of views. (Brisque/NIQE)

| Methods | 3 Views | 4 Views | 5 Views | 6 Views | 7 Views | 8 Views |
|---------|---------|---------|---------|---------|---------|---------|
| TFI+D3DGS [29] | 83.4/23.87 | 73.6/21.67 | 65.2/20.81 | 62.9/18.60 | 59.8/16.20 | 56.53/15.34 |
| TFI+STG [26] | 71.0/22.80 | 61.2/17.50 | 57.0/16.70 | 52.1/14.60 | 42.0/13.10 | 36.45/10.53 |
| TFI+4DGS [45] | 70.3/22.70 | 60.8/18.40 | 56.4/16.50 | 51.8/14.40 | 43.6/13.90 | 37.57/9.93 |
| **Ours** | **63.43/18.54** | **53.62/15.67** | **45.23/13.81** | **32.91/11.67** | **29.84/10.20** | **26.86/9.86** |

in both visual quality and quantitative score. The quantitative comparisons across different methods are summarized in Table 5. It proves that more view numbers will lead to higher rendering quality. This means that our method could be extended to more views in the future.

**The effect of MSDI's Generator.** To validate the effectiveness of our MSDI module for spike2image initialization, we conducted quantitative comparisons with representative spike-to-image methods, including TFI, TFP, and spk2img. On the synthetic dataset where ground truth images are available, we evaluated the spike-to-img quality using average PSNR and SSIM. The results are summarized in Tab 6. Note the reconstruction images in test have the same view with the training images, hence the improvements are more clearly than NVS.

Table 6: Quantitative comparison on average quality.

| Metric | TFI [3] | TFP [3] | TVS [4] | Spike2img [5] | MSDI's images |
|--------|---------|---------|---------|---------------|---------------|
| Avg PSNR↑ | 24.51 | 25.37 | 22.43 | 30.76 | **34.90** |
| Avg SSIM↑ | 0.850 | 0.852 | 0.821 | 0.904 | **0.961** |

**The effect of Whole MSDI.** This part provides a more detailed evaluation of the module MSDI. Specifically, we feed the MSDI outputs (i.e., reconstructed images, point clouds, and poses) into the standard 4DGS procedure, respectively, to validate the contribution of MSDI. In addition, the traditional spike-to-image methods (e.g., TFI) and MSDI+SPSS are also listed for comparison. The experiments are conducted on the synthetic dataset, and the average performances are reported in the following Tab. 7. It indicates MSDI's contribution positively.

Table 7: Focus Comparison of MSDI without SPSS.

| Method | Avg PSNR↑ | Avg SSIM↑ | Avg LPIPS↓ |
|--------|-----------|-----------|------------|
| TFI + COLMAP + 4DGS | 21.08 | 0.849 | 0.253 |
| TFI + DUST3R + 4DGS | 25.66 | 0.893 | 0.205 |
| MSDI IMAGE + COLMAP + 4DGS | 22.43 | 0.854 | 0.233 |
| MSDI IMAGE + DUST3R + 4DGS | 25.76 | 0.898 | 0.203 |
| MSDI + 4DGS | 26.66 | 0.903 | 0.198 |
| MSDI + SPSS (Ours) | **27.74** | **0.912** | **0.190** |

## 5 Conclusions

This paper introduces Spike4DGS, a novel framework that seamlessly integrates multiple spike streams captured by a spike array into 4DGS training, effectively addressing the challenges of reconstructing high-speed dynamic scenes. Spike4DGS designs a novel Multi-view Spike-based Dense Initialization module to obtain dense point clouds from continuous spike streams and a Spike-Pixel Synergy Supervision strategy to optimize the parameters for enhanced rendering quality. We contribute two novel datasets and conduct comprehensive evaluations. The results on the datasets demonstrate that our Spike4DGS surpasses previous SOTA dynamic reconstruction approaches in high-speed dynamic scenes, with almost no sacrifice in training cost and rendering FPS.

**Limitation.** While our spike camera array is functional, the approach has limitations which we will solve in our future works. The images rendered by our Spike4DGS are grayscale due to the lack of RGB information. Additionally, the spike array is not portable, which limits its use in mobile or field-based applications.

**Acknowledgements** The study was funded by the National Natural Science Foundation of China under contracts Shenzhen Science and Technology Program(KQTD20240729102051063), No. 62422602, No. 62425101, No. 62332002, No. 62372010, No. 62027804, No. 62088102, No. 62206281, and the major key project of the Peng Cheng Laboratory (PCL2021A13 and PCL2024AS204). Computing support was provided by Pengcheng Cloudbrain.

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
