# OpenReview forum: "Spike4DGS: Towards High-Speed Dynamic Scene Rendering with 4D Gaussian Splatting via a Spike Camera Array"
_NeurIPS.cc/2025/Conference — NeurIPS 2025 poster_

### Official Review · Reviewer_TNTX · 2025-07-01

**Clarity:** 2
**Significance:** 3
**Originality:** 2
**Rating:** 4
**Confidence:** 4

**Summary:**

This paper proposes using a multi-view spike camera array to synthesize novel views of dynamic scenes. The authors first construct both synthetic and real-world multi-view spike datasets and introduce a GS initialization module that directly estimates dense point clouds and camera poses from spike streams, without relying on existing SfM tools like COLMAP or DUST3R. The framework further integrates 4D Gaussian Splatting for high-speed dynamic scene modeling.

**Questions:**

How does the spike-based dense initialization compare to initialization based on original images (e.g., COLMAP)? Table 3 only compares against spike-reconstructed images, which are of low quality.

While some inaccuracy in point cloud is acceptable for initialization, it would be informative to provide some evaluations of point clouds and camera poses, especially compared to COLMAP and DUSt3R.

When using MSDI for initialization, are the poses further refined in 4DGS stage? Is the pose accuracy comparable to COLMAP-based pipelines?

What is the computational cost of the proposed method?

**Ethical Concerns:**

["NO or VERY MINOR ethics concerns only"]

**Final Justification:**

Based on the first rebuttal, I am willing to raise the score to borderline accept. However, I am not convinced by the author's subsequent clarifications, particularly as I believe the amount of spatiotemporal data of spikes significantly exceeds that of sharp image frames, so I am unable to give a more positive rating.

**Limitations:**

yes

**Quality:**

2

**Strengths And Weaknesses:**

Strengths

A new dataset with multi-view spike camera data covering both synthetic and real-world scenes is introduced, which is valuable for further research.

Compared with traditional two-stage methods that first reconstruct images before applying 4DGS, the proposed approach shows better performance, demonstrating the benefit of joint modeling.

Weaknesses

Although spike cameras offer high temporal resolution, this advantage is neither fully exploited in the framework design nor clearly reflected in the experimental results.

On the synthetic dataset shown in Figure 3, all methods achieve suboptimal results. It raises the question of whether image-based NVS methods (e.g., COLMAP + 4DGS) suffer from similar artifacts on this data.

The authors claim that traditional methods rely on high-quality image reconstruction (Line 140), but the proposed method also requires reconstructed images (Line 170), and the deformed Gaussians in Section 3.4 are highly dependent on image quality. The spike signals only serve as auxiliary loss.

While several spike-based NVS methods are mentioned in Section 2.2, they are not directly compared in experiments. Can those methods be adapted to the multi-view setting? This deserves further discussion.

The proposed multi-view spike-based dense initialization lacks clarity in what makes it unique to spike data. If the input were replaced with multi-view images, the module might still function.

It remains unclear what the core advantage over the "spike-to-image + image 4D NVS" pipeline is: better initial point clouds and poses? Higher-quality reconstructed images? Or improved supervision through spike constraints?

---

> ### Author Rebuttal · Authors · 2025-07-30
>
> # Response to W1
> >Although spike cameras offer high temporal resolution, this advantage is neither fully exploited in the framework design nor clearly reflected in the experimental results.
>
> Thanks for the reviewer’s insightful comment on underusing the spike camera’s high temporal resolution. To address this, we performed an ablation study by downsampling the spike data from 20,000 Hz to lower temporal resolutions while keeping the rest unchanged. This was done by removing frames at fixed intervals to simulate lower frame rates. Results are shown in Table.R1 on our synthesis dataset.
>
> #### **Table.R1 Performance of Spike4DGS under Varying Temporal Resolutions of Spike Input.**
> |Hz|20000|10000|5000|2000|1000|500|200|100|
> |--|--|--|--|--|--|--|--|--|
> |PSNR↑|**27.74**|25.43|22.71|21.88|18.32|16.87|16.22|15.54|
> |SSIM↑|**0.912**|0.885|0.842|0.810|0.745|0.685|0.650|0.625|
>
> The reconstruction quality degrades significantly as the temporal resolution decreases. These results demonstrate that our method implicitly leverages and benefits from the high temporal resolution provided by the spike camera.
>
> # Response to Q1 and W2
> >How does the spike-based dense initialization compare to initialization based on original images (e.g., COLMAP)? Table 3 only compares against spike-reconstructed images, which are of low quality.
>
> >On the synthetic dataset shown in Figure 3, all methods achieve suboptimal results. It raises the question of whether image-based NVS methods (e.g., COLMAP + 4DGS) suffer from similar artifacts on this data.
>
> Thank you for pointing this out. In response to your question, we conducted additional experiments on synthetic data. Specifically,  two types of images are extracted from the CARLA simulator:
> 1) CI(Clean images): The clean RGB images without  motion blur, which is an ideal case to validate the upper limit of RGB-based methods;
> 2) BI(Blurred images): The RGB images with motion blur, which are closer to realistic situations.
>
>  Based on CI and BI, we experiment on the synthetic dataset, and the average performances are reported. Since Dust3R often has better results than COLMAP, Dust3R is used for the initialization of CI and BI. For fair comparison, all rendering images are evaluated in gray style.  The results on the synthesis dataset are shown in Table.R2.
>
> #### **Table.R2 3D Reconstruction Performance Comparison from Different Input Data**
>
> |Method|PSNR↑|SSIM↑|
> |--|--|--|
> |CI+Dust3R+4DGS(clean)|30.31|0.942|
> |BI+Dust3R+4DGS(blur)|22.72|0.861|
> |Ours Spike4DGS|27.74|0.912|
>
> The results show that, in an ideal case where the motion blur doesn't exist, the clean RGB image could achieve good performance. However, considering the realistic case with motion blur, the performance of the RGB-based method is limited. This is also our main motivation to introduce spike cameras.
>
> We will add this statement in the revision.
>
> # Response to W3
> >The authors claim that traditional methods rely on high-quality image reconstruction (Line 140), but the proposed method also requires reconstructed images (Line 170), and the deformed Gaussians in Section 3.4 are highly dependent on image quality. The spike signals only serve as auxiliary loss.
>
> We acknowledge that our previous wording may have been ambiguous.
>
> For the statement of Line 140 and Line 170:  The "traditional methods" mentioned in Line 140 refer specifically to image-based initialization frameworks such as COLMAP or Dust3R. These methods require generating images first, and then performing COLMAP or Dust3R on the images; hence, the quality of image reconstruction directly influences the point clouds and camera poses. In contrast, our MSDI predicts the image, point clouds, and camera poses end-to-end. Hence, the quality of image reconstruction wouldn't influence the point clouds and camera poses.
>
> For the statement of Section 3.4:  Our method is indeed dependent on image quality, but the loss of spike signals also plays an important role in SPSS. Spike signals are high-temporal-resolution streams that can capture subtle dynamic changes in the scene, such as motion edges and rapid variations. Our spike total variation loss(TV loss) in SPSS helps keep motion smooth and consistent over time. As shown in Table.R3, we validate the effectiveness of the spikeloss through empirical evaluation:
>
> #### **Table.R3 the Effectiveness of the Spikeloss**
> |Supervision|PSNR↑|SSIM↑|
> |------|-----|-----|
> |Ours spike TV loss|**27.74**|**0.912**|
> |w/o TV loss, use L1 spikeloss|26.87|0.906|
> |w/o any spikeloss|26.66|0.903|
>
> # Response to Q2
> > While some inaccuracy in point cloud is acceptable for initialization, it would be informative to provide some evaluations of point clouds and camera poses, especially compared to COLMAP and DUSt3R.
>
> Thank you for your suggestion for the additional experience. We quantitatively evaluate the camera pose estimation accuracy of MSDI in comparison with Colmap and Dust3R[1] on the synthesis dataset. The ground truth camera poses could be obtained from the Carla simulator. The results are shown in the table.R4 below:
>
> #### **Table.R4 Comparison of Camera Pose Estimation Results.**
> |Method|Rotation Error (°) ↓|Translation Error (%) ↓
> |---|---|---|
> |TFI+COLMAP|2.72|9.8|
> |TFI+Dust3R|1.27|3.5|
> |MSID IMAGE+COLMAP|2.62|9.6|
> |MSDI IMAGE+Dust3R|1.19|3.2|
> |Ours (MSDI)|**0.94**|**2.1**|
>
> This demonstrates that even when using the same spike data, our MSDI method outperforms the TFI + image-based initialization framework, as MSDI is not limited by image quality.
>
> # Response to Q3
> >When using MSDI for initialization, are the poses further refined in 4DGS stage? Is the pose accuracy comparable to COLMAP-based pipelines?
>
> To save computational resources, we did not further optimize the camera poses during the 4DGS stage as same as original. This is because our initialization via MSDI already yields higher pose accuracy compared to COLMAP-based pipelines from the results in Table.R4.
>
> # Response to Q4
> >What is the computational cost of the proposed method?
>
> The computational cost of our method consists of two main parts:
>
> 1) The MSDI module is trained using 4 NVIDIA RTX A6000 GPUs in parallel and takes approximately 40 hours. Its inference usually takes about 10s, compared with several minutes in Colmap.
> 2) The subsequent reconstruction stage runs on a single RTX 4090 GPU and takes around 20+ minutes, depending on the number of frames and views. The time is a little longer than the original 4DGS.
>
> # Response to W4
> >While several spike-based NVS methods are mentioned in Section 2.2, they are not directly compared in experiments. Can those methods be adapted to the multi-view setting? This deserves further discussion.
>
> Thank you for the valuable comment. The methods mentioned in Section 2.2 are indeed designed for static scene reconstruction, even though they also utilize neuromorphic (spike) cameras. To the best of our knowledge, our work is the first to extend neuromorphic vision into the dynamic 4D reconstruction setting. **These static methods are adapted to the multi-view setting but are not directly applicable to dynamic scenes.** Thus, our proposed framework fills this gap by enabling temporally consistent 4D reconstruction from asynchronous spike data in dynamic environments.
>
> Following your suggestions, we conducted additional experiments. We manually integrated a deformation network from 4DGS into SpikeGS and SpikeNeRF (denoted as Dy-SpikeGS and Dy-SpikeNeRF).  Experiments are conducted on the synthesis dataset, and the average performances are reported in Table R5.  As shown in Table R5, both original and modified dynamic versions of SpikeGS and SpikeNeRF have poor performance. **Note Dy-SpikeGS and Dy-SpikeNeRF are both re-designed and re-implemented, their results are just for reference.**
>
> #### **Table.R5 Comparison with manually Modified SpikeGS and SpikeNeRF.**
>
> | Method        |Avg PSNR(dB)↑ |Avg SSIM↑
> |---------------|-----------|-------|
> | Original SpikeNeRF| 14.43|0.576
> | Original SpikeGS | 20.03|0.808
> | Dy-SpikeNeRF| 16.70     | 0.628
> | Dy-SpikeGS | 23.31     | 0.835
> | Spike4DGS   | **27.74**     | **0.912**
>
> # Response to W5
> >The proposed multi-view spike-based dense initialization lacks clarity in what makes it unique to spike data. If the input were replaced with multi-view images, the module might still function.
> The MSDI is a framework tailored for spike-based 3D vision, which is trained by spike data. So the deal input of the module is spike data, not images like Colmap's input. The module couldn't function if the input were replaced with multi-view images.
>
> # Response to W6
> >It remains unclear what the core advantage over the "spike-to-image + image 4D NVS" pipeline is: better initial point clouds and poses? Higher-quality reconstructed images? Or improved supervision through spike constraints?
>
> We first compare the MSDI images with some spike-to-image methods, including TFI, TFP, and spk2img. On the synthesis dataset, where ground truth images are available, we evaluated the spike-to-image quality using PSNR and SSIM. The results are summarized in Table.R6.
>
> #### **Table.R6 Spike-to-image Quality**
> |Metric|TFI|TFP|TVS|Spike2img|MSDI'simages|
> |------|------|------|------|-------------|-------------|
> |Avg PSNR|24.51|25.37|22.43|30.76|**34.90**|
> |Avg SSIM|0.850|0.852|0.821|0.904|**0.961**|
>
> Based on the results in Table.R6, we acknowledge that a little of the final improvement comes from the enhanced image quality produced by MSDI, but not all. We conducted an additional ablation study in Table.R7:
>
> **Table.R7. Ablation on Each Module**
> |Method|PSNR↑|SSIM↑|
> |------|-----|-----|
> |TFI+DUST3R+4DGS|25.66|0.893|
> |MSDI Image+DUST3R+4DGS|25.76|0.898|
> |MSDI(Image and pointclouds/poses)+4DGS|26.66|0.903|
> |MSDI(Image and pointclouds/poses)+4DGS+SPSS|27.74|0.912|
>
> It can be observed that point clouds/poses and images from MSDI all improve performance, and the SPSS module further enhances the utilization of spike data, leading to better results.

---

> > ### Comment · Reviewer_TNTX · 2025-08-02
> >
> > Thank you to the author for the detailed response, which has addressed some of my concerns. However, I still have a few questions.
> >
> > Removing spike frames would certainly degrade performance, as a significant amount of information would be lost. I am concerned about the advantages of the high temporal resolution of spike cameras over images for NVS, and the unique design of the framework in this paper that leverages this characteristic of spike cameras should be explicitly pointed out. The issue of image blurring, which the author mentioned, should be added to the discussion in the revised paper. As for my latter concern about the unique design, I don't think it has been addressed yet. Regarding my concern about the MSDI module, I think such a design could be retrained for image data. The authors did not highlight the innovative aspect of this module beyond inputting spike data.
> >
> > Results based on spike data are worse than those based on sharp images, and this is worth further discussion. From a data acquisition perspective, spike cameras do not have motion blur and capture much more information than low-frame-rate sharp images. Therefore, better results based on spike data should be expected.
> >
> > Regarding the Response to W4, although [29] has not yet been open-sourced, further discussion is needed beyond simple citation.
> > [29] Deformable Neural Radiance Fields using RGB and Event Cameras, ICCV 2023

---

> > > ### Author Response · Authors · 2025-08-04
> > > **Further Responses**
> > >
> > > We sincerely thank the reviewers for their further questions. Below are our responses to these new questions.
> > >
> > > # Response to Q1
> > >
> > > ## (1) The unique design of spike cameras:
> > >
> > > Our method includes specific designs for the high temporal resolution characteristics of spike data. These are mainly reflected in the spike loss in SPSS. Besides, MSDI is a processing framework designed for spike data initialization.
> > >
> > > **Spikeloss**: Firstly, a spike variation loss is proposed in SPSS to effectively capture the fine-grained and complex motion in spike streams. Unlike naive L1 losses, our loss explicitly supervises temporal variations by considering both the mean and variance over time. This encourages local temporal smoothness while reducing noise, which is crucial for accurately modeling the temporal consistency of spike signals. As shown in the previous rebuttal Table R.3, compared to the naive L1 loss, our spike TV loss achieves significantly higher PSNR and SSIM, surpassing the baseline and establishing a new SOTA.
> > >
> > > **MSDI**: Secondly, the MSDI is trained jointly and could generate images and estimate poses jointly. It outperforms on 2-stage based pose estimation as shown in the previous rebuttal Table R.4.  We have additionally provided results where image generator and pose estimation are trained separately as below in the synthesized data.  It indicates that the joint learning not only reduces the computation cost but also leverages their complementarity.
> > > ***
> > > #### **Pose Estimation**
> > > |Training|Rot. Err. (°)↓|Trans. Err. (%)↓|
> > > |---|---|---|
> > > |Joint MSDI|**0.94**|**2.1**|
> > > |Separate MSDI|0.98|2.3|
> > > ***
> > > #### **Image Generator**
> > > |Metric|Spike2img|separate MSDI|joint MSDI|
> > > |---|---|---|---|
> > > |Avg PSNR|30.76|30.12|**34.90**|
> > > |Avg SSIM|0.904|0.911|**0.961**|
> > > ***
> > > ## (2) The contribution of MSDI:
> > >  In model design, we combine the similar network used in spike processing literature [1] and image-based Dust3R [2]. The key contribution is our design of training strategy, and it could handle the challenge of the lack of  spike-based training datasets for structure from motion (SfM).
> > >
> > > As stated in [2], training Dust3R needs 8.5M image-GT pairs from various scenes. However, the spike camera haven’t been widely used, and the labelled training data is hard to collect. Although the simulation data from Carla is available, the domain is limited and cannot generalize to the real scene effectively. Hence, we try to transfer the prior knowledge of pre-trained Dust3R from the image domain to the spike domain, and initialize the ViT backbone in MSDI with pre-trained weights from DUST3R. As shown below on the synthesis and real datasets, MSDI without Dust3R initialization yields poor results.
> > > ***
> > > |Method|PSNR↑|SSIM↑|
> > > |---|---|---|
> > > |MSDI w/o pretrainI+SPSS|20.41|0.801|
> > > |Ours|**27.74**|**0.912**|
> > > ***
> > > |Method|Brisque↓|NIQE↓|
> > > |---|---|---|
> > > |MSDI w/o pretrain+SPSS|53.21|16.32|
> > > |Ours|**28.63**|**9.05**|
> > > ***
> > >
> > >  # Response to Q2
> > >
> > > Although our results are slightly lower than those using sharp images, the latter should be viewed as a theoretical upper bound of RGB-based methods rather than a fair real-world baseline. Since these sharp images are idealized outputs from the CARLA simulator which is based on the Unreal Engine. The images are noise-free and captured at virtually infinite frame rates. In real-world scenarios, RGB cameras are limited by frame rate, and fast motion inevitably causes severe motion blur . As a result, reconstructions from such ideal images serve only as a reference for the best possible performance of RGB-based methods.
> > >
> > > # Response to Q3
> > >
> > > We thank the reviewer for suggesting this related work. Upon careful consideration, we acknowledge that the task discussed in the referenced paper is closely related to ours. However, there are key differences between the two approaches, and [3] is not open-sourced yet, which makes a direct comparison challenging. We will include [3] and clarify the difference in revision:
> > >
> > > For the NVS task, [3] captures dynamic scenes using a moving monocular camera which could produce both RGB and DVS. Hence, they could estimate trajectories of DVS-based camera poses by predicting residual poses initialized from RGB images. In contrast, we use a fixed multi-camera array to record high-speed scenes. Therefore, we propose MSDI, which can map Spike data to both camera poses and point cloud data directly.
> > >
> > > During reconstruction, [3] uses the RGB images to capture the global structure of the radiance field, and then enhance the structures in space and time by using events. In contrast, our method uses only Spike data throughout the reconstruction pipeline. In addition, [3] is based on Nerf and our method is designed on 4DGS, which are two totally different baselines.
> > >
> > > ***
> > > [1] Self-supervised mutual learning for dynamic scene reconstruction of spiking camera. IJCAI 2022.
> > >
> > > [2] Dust3r: Geometric 3d vision made easy. CPVR 2024
> > >
> > > [3] Deformable Neural Radiance Fields using RGB and Event Cameras. ICCV 2023

---

> > > > ### Comment · Reviewer_TNTX · 2025-08-05
> > > >
> > > > Thank you for your further response. Based on the first rebuttal, I am willing to raise the score to borderline accept. However, I am not convinced by the author's subsequent clarifications, particularly as I believe the amount of spatiotemporal data of spikes significantly exceeds that of sharp image frames, so I am unable to give a more positive rating.

---

> > > > > ### Author Response · Authors · 2025-08-05
> > > > > **Response**
> > > > >
> > > > > We sincerely thank the reviewer for these thoughtful comments and for raising the rating to borderline accept. We will incorporate these discussions and results into the final version and consider these comments in our future work.

---

### Official Review · Reviewer_mUPw · 2025-07-02

**Clarity:** 2
**Significance:** 3
**Originality:** 3
**Rating:** 4
**Confidence:** 4

**Summary:**

This paper constructs an array of spike cameras and uses it to capture multiview images of a dynamic scene. With this novel dataset in hand, the authors then develop and apply 4D Gaussian splatting reconstruction framework to reconstruct a dynamic volumetric reconstruction of the scene. The proposed reconstruction algorithm has a novel initialization scheme and loss function. The initialization scheme estimates camera poses and an initial point cloud from an input spike stream using a vision transfomer. The loss function consists of (1) a spike loss between the capture data and spikes rendered by the reconstruction and (2) a pixel loss between images reconstructed directly from the spike cameras and those rendered by the reconstruction. The proposed method outperforms existing methods on real and simulated data.

**Questions:**

Could the authors provide qualitative evidence that the method is successfully reconstructing 3D geometry? E.g., a visualization of the 3DGS depth map or views rendered from a sequence of perspectives that were not originally captured. This is my major criticism of the current manuscript---it claims 3D reconstruction but doesn't convincingly demonstrate it.

Are Figures 3, 4, and 5 rendered from novel viewpoints or viewpoints that were already captured? How close are the rendered views' perspectives to an existing view's?

Will the datasets be publicly released? The paper lists the dataset as a contribution, suggesting it will be, but this should be made more explicit.

Did the cameras have overlapping view points? (In the main text they are positioned s.t. their views likely wouldn't overlap.) If the viewpoints are not overlapping and the camera is not moving, how is 3D reconstruction being performed?

**Ethical Concerns:**

["NO or VERY MINOR ethics concerns only"]

**Final Justification:**

The authors clarified that their reported results were actually testing novel view synthesis capabilities.

**Limitations:**

Yes

**Quality:**

3

**Strengths And Weaknesses:**

## Strengths:

To my knowledge, the proposed method and camera array are novel

The dataset, assuming it is publicly released, would be useful

The proposed method significantly outperforms existing methods.

## Weaknesses:

Missing Evidence: The paper provides limited evidence that 3D reconstruction was successful. While tables 2-4 in the supplement report novel view reconstruction accuracy on simulated data, there are no "fly-through" in the supplement, depth maps, or other qualitative evidence showing that accurate 3D geometry was recovered. The demo videos are rendered from a fixed perspective and fail to demonstrate any 3D geometry was successfully recovered. It's unclear if the results shown in the paper are rendered from a novel view point or one that was already captured by one of the cameras.

Clarity and missing information: The main text shows the cameras positioned in such as way that their viewpoints wouldn't overlap and 3D reconstruction would be extremely challenging. The supplement suggests these cameras were positioned with overlapping views. Please provide information about the position, orientation, FoV, etc. of each camera.

In the abstract and title, the paper uses the word "rendering" when what is really meant is "reconstruction".

---

> ### Author Rebuttal · Authors · 2025-07-29
>
> Thank you for your positive and thoughtful comments. We would like to address your concerns and answer your questions in the following.
>
> # Response to Q1 and W1
> >Could the authors provide qualitative evidence that the method is successfully reconstructing 3D geometry? E.g., a visualization of the 3DGS depth map or views rendered from a sequence of perspectives that were not originally captured. This is my major criticism of the current manuscript---it claims 3D reconstruction but doesn't convincingly demonstrate it.
>
> >The paper provides limited evidence that 3D reconstruction was successful. While tables 2-4 in the supplement report novel view reconstruction accuracy on simulated data, there are no "fly-through" in the supplement, depth maps, or other qualitative evidence showing that accurate 3D geometry was recovered. The demo videos are rendered from a fixed perspective and fail to demonstrate any 3D geometry was successfully recovered. It's unclear if the results shown in the paper are rendered from a novel view point or one that was already captured by one of the cameras.
>
> We have conducted fly-through visualizations to demonstrate the 3D stereo effect. Due to the rebuttal policy, we are currently unable to provide videos here. Upon public release, we will provide videos showcasing this stereo visualization. We appreciate the reviewer highlighting the importance of this aspect.
>
> To further verify the reconstruction capability of our method, we have computed depth maps from our model's outputs. We extracted depth information from our synthesis dataset as ground-truth and used rendering depth images from our methods and baselines to perform quantitative comparisons. The results reported in Table.R1 are averaged across five different scenes from our synthetic dataset. We follow two metrics (Abs Rel and δ < 1.25) of Dust3R[1] to evaluate the depth results.
>
> #### **Table.R1 Evaluation of depth estimation accuracy on Avg Abs Rel and δ < 1.25.**
> | Method              |Avg Abs Rel ↓ |Avg δ < 1.25 ↑ |
> |---------------------|-----------|------------|
> | Spike2img + D3DGS   | 0.523     | 0.512     |
> | Spike2img + STG     | 0.340     | 0.538      |
> | Spike2img + 4DGS    | 0.342     | 0.536      |
> | Our Spike4DGS  | **0.195** | **0.565**  |
>
> Abs Rel (Average Absolute Relative Error) quantifies the mean absolute discrepancy between rendering and ground truth depth values, normalized by the ground truth depth. Lower Abs Rel values correspond to higher accuracy, indicating that the predicted depths closely approximate the true depths relative to their magnitude.
>
> δ < 1.25 (Threshold Accuracy at 1.25) denotes the fraction of rendering depth estimates for which the ratio between prediction and ground truth lies within the threshold of 1.25. A higher δ value means a greater proportion of rendering depth falling within an acceptable error margin, reflecting superior depth estimation fidelity.
>
> # Response to Q2
>
> > Q2: Are Figures 3, 4, and 5 rendered from novel viewpoints or viewpoints that were already captured? How close are the rendered views' perspectives to an existing view's?
>
> We apologize for the ambiguity. In fact, for both the real-world and synthetic datasets, the images shown in our main text Figures 3, 4, and 5 comparisons are rendered from **a novel view**. Specifically, for the real-world dataset, we use a 3×3 spike camera array and select the **central view** as the novel test view for evaluation. Similarly, for the synthetic dataset, we simulate the same 3×3 camera grid and also select the **central view** as the test viewpoint. All experiments in our main paper follow this setting. **We did not use training views for testing, strictly following the experimental protocol of the Novel View Synthetic task.**
>
> # Response to Q3
>
> >Will the datasets be publicly released? The paper lists the dataset as a contribution, suggesting it will be, but this should be made more explicit.
>
> We appreciate the reviewer’s attention to our dataset contribution.
> For the real-world dataset, we use purely spike data from 9 views. For the synthesis dataset, we additionally include clean RGB images and RGB images with motion blur from corresponding views. These RGB images are incorporated specifically to support evaluation metrics and qualitative comparisons.
>
> Due to the double-blind review policy, we are currently unable to provide direct dataset links. We confirm that the dataset, along with all related resources, will be made publicly available with the publication of the paper.
>
> # Response to W2  and Q4
> > Clarity and missing information: The main text shows the cameras positioned in such as way that their viewpoints wouldn't overlap and 3D reconstruction would be extremely challenging. The supplement suggests these cameras were positioned with overlapping views. Please provide information about the position, orientation, FoV, etc. of each camera.
>
> >Did the cameras have overlapping view points? (In the main text they are positioned s.t. their views likely wouldn't overlap.) If the viewpoints are not overlapping and the camera is not moving, how is 3D reconstruction being performed?
>
> The camera arrangement illustrated in Figure 2 of the main text is for demonstration purposes only and does not exactly represent the actual setup used during data collection.
>
> **For our real-world dataset**, we used a 3×3 camera array from a square layout and ensured that the object was centered directly in front of the middle camera. The distance between neighboring cameras' positions is around 20 cm. During data collection, all surrounding cameras were tilted inward and oriented toward the center camera, forming a conical viewing region with significant overlap among the fields of view. This configuration ensures that each scene is well covered from multiple angles, which is reasonable for successful 3D reconstruction.
>
> # Response to W3
> >In the abstract and title, the paper uses the word "rendering" when what is really meant is "reconstruction".
>
> Thank you for pointing out the misuse of the term "rendering" in the abstract and title. The  "reconstruction"  and  "rendering"  corresponds to the 3D scene and novel view synthesis, respectively. We will carefully check the manuscript  to avoid any confusion.
>
> ***
> #### [1]Shuzhe Wang, Vincent Leroy, Yohann Cabon, Boris Chidlovskii, and Jerome Revaud. Dust3r: Geometric 3d vision made easy. In CVPR, 2024.

---

> > ### Comment · Reviewer_mUPw · 2025-08-01
> >
> > Thank you for your thorough response. Could you elaborate a little on the response to Q2. You state "we use a 3×3 spike camera array and select the central view as the novel test view for evaluation". Does that mean you supervised on the 8 views around the periphery of the array and tested on the central one? Or was the central view also used for supervision?

---

> > > ### Author Response · Authors · 2025-08-02
> > > **Response**
> > >
> > > Thank you for your further official comment!
> > >
> > > As you correctly inferred,  the center view is strictly held out as the test view and is not used for supervision during training. Specifically, we supervise the model using the surrounding 8 views on the periphery of the array, while the center view is entirely excluded from the training process. This setup allows us to rigorously evaluate the model’s ability to generalize to unseen viewpoints. The center view is chosen for testing because it typically provides a more complete and challenging view of the scene, thereby serving as a reliable indicator of reconstruction quality.

---

> > > > ### Comment · Reviewer_mUPw · 2025-08-06
> > > >
> > > > Thanks. I've updated my score.

---

> > > > > ### Author Response · Authors · 2025-08-07
> > > > > **Response**
> > > > >
> > > > > I sincerely appreciate your thoughtful comments and your willingness to update the score. Thank you for your time and effort！

---

### Official Review · Reviewer_Q7d2 · 2025-07-03

**Clarity:** 3
**Significance:** 2
**Originality:** 3
**Rating:** 4
**Confidence:** 3

**Summary:**

The paper proposes Spike4DGS, the first method for reconstructing high-speed dynamic scenes using a spike camera array. The approach begins with a Multi-view Spike-based dense initialization to generate an initial point cloud, estimate camera poses, and synthesize RGB images from the spike streams. These synthesized images and estimated poses are then used to optimize 4D Gaussian Splatting (4DGS). To further enhance reconstruction quality, the method incorporates a novel Spike Variation Supervision mechanism that leverages the unique characteristics of spike data. Additionally, the authors create two benchmark datasets, one synthetic and one real-world, to evaluate the method’s performance. Extensive experiments demonstrate that Spike4DGS achieves superior results compared to existing baselines.

**Questions:**

See weakness 1, 2, 3.

**Ethical Concerns:**

["Major Concern: Data privacy, copyright, and consent"]

**Final Justification:**

The authors have addressed my concerns regarding the comparison with Dynamic Spike-GS and Dynamic Spike-NeRF, the quality comparison between the proposed MSDI and traditional spike-to-image methods, as well as the clarification of the training view setting. Thus I will maintain my positive score.

**Limitations:**

Discussed in supplmentary section I.

**Quality:**

2

**Strengths And Weaknesses:**

Strength
1. The paper introduces Multi-view Spike-based Dense Initialization (MSDI) to jointly initialize the point cloud and estimate camera poses directly from spike streams, while simultaneously recovering images. This unified approach replaces the traditional two-step pipeline: first recovering images and then performing point cloud initialization, thereby reducing reliance on spike-to-image conversion methods, which may produce suboptimal results.
2. The authors introduce one synthetic and one real-world dataset as benchmarks for evaluating high-speed dynamic scene reconstruction from spike streams. These datasets may serve as valuable resources for the community and help advance future research in spike-based reconstruction.
3. A spike variation loss is introduced to exploit the rich texture and motion cues inherent in the spike stream, thereby enhancing the quality of the reconstructed scenes.
4. The paper is well-written and easy to follow.


Weakness
1. All baseline methods follow a two-stage pipeline of spike-to-image conversion followed by dynamic reconstruction. Would it be more appropriate to consider a stronger baseline by directly modifying existing spike-based methods such as SpikeGS or SpikeNeRF to support dynamic rendering—e.g., by integrating a deformation field? This could provide a more direct and fair comparison.
2. A visual and quantitative comparison between the RGB images generated by spike-to-image methods (e.g., TFI, TFP, spk2img) and those produced by the proposed MSDI module would better validate the effectiveness of the spike-based initialization.
3. In Figure 5, the rendering quality degrades noticeably as the number of input views decreases. It would be helpful for the authors to clarify how many input views were used in the comparisons with baseline methods. Additionally, providing a comparison with baselines under varying numbers of input views would offer valuable insight into the robustness of the proposed method in sparse-view settings.

---

> ### Author Rebuttal · Authors · 2025-07-28
>
> # Response to W1
> >All baseline methods follow a two-stage pipeline of spike-to-image conversion followed by dynamic reconstruction. Would it be more appropriate to consider a stronger baseline by directly modifying existing spike-based methods such as SpikeGS or SpikeNeRF to support dynamic rendering—e.g., by integrating a deformation field? This could provide a more direct and fair comparison.
>
> We thank the reviewer for his insightful suggestion. **Both SpikeGS [1] and SpikeNeRF [2] focus on reconstructing static scenes and are not applicable to dynamic scenarios involving object motion.**
>
> Following your suggestions, we conducted additional experiments. As there are currently no publicly available dynamic extensions for these methods, we manually integrated a deformation network from 4DGS into SpikeGS and SpikeNeRF (denoted as Dy-SpikeGS and Dy-SpikeNeRF) and adapted their data reader pipelines accordingly.  Experiments are conducted on the synthesis dataset, and the average performances are reported in Table R1.  As shown in Table R1, both original and modified dynamic versions of SpikeGS and SpikeNeRF have poor performance. **Note Dy-SpikeGS and Dy-SpikeNeRF are both re-designed and re-implemented, their results are just for reference.**
>
> #### **Table.R1 Comparison with manually Modified SpikeGS and SpikeNeRF.**
>
> | Method        |Avg PSNR(dB)↑ |Avg SSIM↑
> |---------------|-----------|-------|
> | Original SpikeNeRF| 14.43|0.576
> | Original SpikeGS | 20.03|0.808
> | Dy-SpikeNeRF| 16.70     | 0.628
> | Dy-SpikeGS | 23.31     | 0.835
> | Spike4DGS   | **27.74**     | **0.912**
>
>
> ***
> # Response to W2
> >A visual and quantitative comparison between the RGB images generated by spike-to-image methods (e.g., TFI, TFP, spk2img) and those produced by the proposed MSDI module would better validate the effectiveness of the spike-based initialization.
>
> To validate the effectiveness of our MSDI module for spike-to-image initialization, **we conducted quantitative comparisons with representative spike-to-image methods, including TFI, TFP, and spk2img.**
>
> On the synthetic dataset where ground truth images are available, we evaluated the spike-to-img quality using average PSNR and SSIM.  The results are summarized in Table.R2. Note the reconstruction images in test have the same view with the training images, hence the improvements are more clearly than NVS.
>
> #### **Table.R2 Comparison of spike-to-img quantitative results on synthesis dataset.**
>
> |   Metric   |  TFI [3] |  TFP[3]  |  TVS[4]  | Spike2img[5] |   MSDI's images   |
> |:------:|:------:|:-----:|:-----:|:---------:|:---------:|
> |Avg  PSNR↑  |  24.51 | 25.37 | 22.43 |   30.76   | **34.90** |
> |Avg  SSIM↑  | 0.850  | 0.852 | 0.821 |   0.904   | **0.961** |
>
> On real-world data, we adopted the no-reference Brisque↓/NIQE↓ metric. The results are summarized in Table.R3. Our MSDI module consistently achieves superior performance across all metrics, demonstrating its ability to produce more accurate and perceptually coherent images from raw spike inputs.
>
> #### **Table.R3 Comparison of spike-to-img Brisque↓/NIQE↓ (↓) on real-world dataset.**
> |   Scene   |  TFI [3] |  TFP[3]  |  TVS[4]  | Spike2img[5] |   MSDI's images   |
> |:---------:|:-----:|:-----:|:-----:|:---------:|:--------:|
> | Avg.Brisque↓ |36.98|33.45|40.73|30.23|20.34
> |Avg.NIQE↓|14.92 |13.31|17.43|10.13|**7.68**
> ***
> # Response to W3
> > In Figure 5, the rendering quality degrades noticeably as the number of input views decreases. It would be helpful for the authors to clarify how many input views were used in the comparisons with baseline methods. Additionally, providing a comparison with baselines under varying numbers of input views would offer valuable insight into the robustness of the proposed method in sparse-view settings.
>
> **Camera Settings**
> In all experiments reported, we used the full 3×3 spike camera array, including 8 training views and 1 testing view, to ensure fair and consistent comparison with baseline methods. The testing view is always the center camera in the array. In Figure 5, the first column corresponds to 4 training views and the test view. The second column corresponds to 6 training views and the test view.
>
> **Comparison Experiments**
> Following your suggestion, we have additionally conducted experiments on baseline methods with varying numbers of training views to provide a more comprehensive comparison. All conditional experiments are conducted on our real-world "Turntable" dataset and shown in Table.R4, and the number of test views is kept to 1 central view.
>
> #### **Table.R4 Impact of Number of Total Training Views for Different Reconstruction Methods.(Brisque↓/NIQE↓)**
> | Methods       | 3 Views       | 4 Views       | 5 Views       | 6 Views       | 7 Views       | 8 Views       |
> |--------------|---------------|---------------|---------------|---------------|---------------|---------------|
> | TFI+D3DGS[6] | 83.4/23.87    | 73.6/21.67    | 65.2/20.81    | 62.9/18.6     | 59.8/16.20    | 56.53/ 15.34   |
> | TFI+STG[6]   | 71.0/22.8     | 61.2/17.5     | 57.0/16.7     | 52.1/14.6     | 42.0/13.1     | 36.45/10.53     |
> | TFI+4DGS[7]  | 70.3/22.7     | 60.8/18.4     | 56.4/16.5     | 51.8/14.4     | 43.6/13.9     | 37.57/9.93     |
> | Ours         | **63.43/18.54** | **53.62/15.67** | **45.23/13.81** | **32.91/11.67** | **29.84/10.2** | **26.86/9.86** |
>
> Our total view configuration experiments follow the same setup as in the main paper, consistently using the center camera of the 3×3 array as the test view. In summary, our method demonstrates strong reconstruction capability and high image quality across different input view densities, maintaining advantages even under sparse view conditions, which reflects the robustness and effectiveness of the approach.
>
> ***
> ##### [1] Jinze Yu, Xin Peng, Zhengda Lu, Laurent Kneip, and Yiqun Wang. Spikegs: Learning 3d gaussian fields from continuous spike stream. In Proceedings of the Asian Conference on Computer Vision, pages 4280–4298, 2024.
> ##### [2] Lin Zhu, Kangmin Jia, Yifan Zhao, Yunshan Qi, Lizhi Wang, and Hua Huang. Spikenerf: Learning neural radiance fields from continuous spike stream. In Proceedings of the IEEE/CVF Conference on Computer Vision and Pattern Recognition, pages 6285–6295, 2024.
> ##### [3] Lin Zhu, Siwei Dong, Tiejun Huang, and Yonghong Tian. A retina-inspired sampling method for visual texture reconstruction. In 2019 IEEE International Conference on Multimedia and Expo (ICME), pages 1432–1437. IEEE, 2019.
> ##### [4] Lin Zhu, Siwei Dong, Jianing Li, Tiejun Huang, and Yonghong Tian. Retina-like visual image reconstruction via spiking neural model. In IEEE Conference on Computer Vision and Pattern Recognition (CVPR), pages 1438–1446, 2020.
> ##### [5] Jing Zhao, Ruiqin Xiong, Hangfan Liu, Jian Zhang, and Tiejun Huang. Spk2imgnet: Learning to reconstruct dynamic scene from continuous spike stream. In Proceedings of the IEEE/CVF Conference on Computer Vision and Pattern Recognition, pages 11996–12005, 2021.
> ##### [6] Zhan Li, Zhang Chen, Zhong Li, and Yi Xu. Spacetime gaussian feature splatting for real-time dynamic view synthesis. In Proceedings of the IEEE/CVF Conference on Computer Vision and Pattern Recognition, pages 8508–8520, 2024.
> ##### [7] Guanjun Wu, Taoran Yi, Jiemin Fang, Lingxi Xie, Xiaopeng Zhang, Wei Wei, Wenyu Liu, Qi Tian, and Xinggang Wang. 4d gaussian splatting for real-time dynamic scene rendering. In Proceedings of the IEEE/CVF Conference on Computer Vision and Pattern Recognition, pages 20310–20320, 2024.

---

> > ### Author Response · Authors · 2025-08-07
> > **Response For Ethical Concerns**
> >
> > We construct both synthetic and real-world datasets for our study.
> >
> > The synthetic dataset is generated using the CARLA simulator and does not involve any real-world individuals or environments.
> >
> > The real-world dataset is collected using a spike camera array in a controlled indoor laboratory setting, without capturing any personally identifiable or sensitive content.
> >
> > All datasets are intended strictly for academic use. We will release them after publication under the Creative Commons Attribution-NonCommercial 4.0 International (CC BY-NC 4.0) license, which supports reproducible research.

---

> > ### Comment · Reviewer_Q7d2 · 2025-08-07
> >
> > Thank you for your detailed response and for providing the additional experimental results.
> >
> > The authors have addressed my concerns regarding the comparison with Dynamic Spike-GS and Dynamic Spike-NeRF, the quality comparison between the proposed MSDI and traditional spike-to-image methods, as well as the clarification of the training view setting.
> >
> > Overall, the rebuttal resolves my concerns. I will wait to see the discussion from other reviewers before deciding whether to raise my score.

---

### Official Review · Reviewer_xVW6 · 2025-07-03

**Clarity:** 2
**Significance:** 3
**Originality:** 3
**Rating:** 4
**Confidence:** 4

**Summary:**

This paper presents a novel framework for high-speed dynamic scene rendering. It uses an array of spike cameras to capture dynamic scenes at very high temporal resolution to reduce motion blur, and then applies 4D Gaussian Splatting (4DGS) for Novel View Synthesis (NVS). The core components include a Multi-view Spike-based Dense Initialization (MSDI) module, which converts spike camera streams into 4DGS readable formats, and a Spike-Pixel Synergy Supervision (SPSS) strategy, which enables direct 4DGS supervision using spike signals.

**Questions:**

The selling points of the paper seem to be the MSDI and SPSS modules, but I am having a hard time assessing their claimed effectiveness. So, my questions are around the method design and evaluation:
1.	Equation (17): Can you explain why you chose the total variation loss rather than an L1 or L2? A brief justification of this design choice would help clarify its role.
2.	MSDI is very interesting to me. It would be helpful if the authors could provide a more detailed evaluation of this module—for example, by feeding the MSDI outputs (point clouds, reconstructed images, and poses) directly into 4DGS without SPSS. This would help identify whether the gains come primarily from better pointclouds/poses (versus DUSt3R) or better images (versus spk2img).
3.	Can you also try replacing only the initialization pointcloud with one from DUSt3R or COLMAP while keeping everything else the same? This would help isolate the contribution of your initialization strategy as any Gaussian Splatting based rendering method is greatly reliant on the initial pointcloud input.
4.	Could you showcase some examples in more realistic settings—such as a car driving down a road in real lighting conditions? I understand that your pipeline assumes uniform lighting, but seeing examples in real-world scenarios would help reviewers better assess its effectiveness and generalizability. I may reconsider my final rating depending on these results.

**Ethical Concerns:**

["NO or VERY MINOR ethics concerns only"]

**Final Justification:**

The author provided an excellent rebuttal that addressed most of my concerns. I would like to raise my rating and suggest that the author add Q3 and Q4 to the revision if the paper is accepted.

**Limitations:**

Yes

**Quality:**

2

**Strengths And Weaknesses:**

Strengths:
1. This paper thoroughly explains the current limitations in dynamic scene rendering, particularly the motion blur introduced by conventional low-speed cameras.
2. The idea of using a spike camera array to alleviate motion blur and enable multi-view NVS is intuitive, reasonable, and novel.
3. The proposed MSDI and SPSS modules, if fully evaluated, have the potential to support many 3DGS-related downstream tasks.
4. Quantitatively, although there is no prior work with an identical setup, the reported metrics surpass the SOTA in dynamic NVS tasks by a significant margin.

Weaknesses:
1. The Methodology section lacks clarity. The authors simply list a series of equations without explaining the intuition or reasoning behind these design choices, making the section very difficult to follow.
2. There is little to no analysis in the Experiments section. It is unclear to the reviewer which components actually contribute to the large performance gains, raising questions about the setup for other methods.
3. Limited generalizability: The real-data experiments only cover small-scale, toy examples. The authors also mention providing ideal lighting conditions when collecting real data. Taken together, these factors raise questions about the pipeline’s ability to handle datasets outside their own collection.
4. Too many typos: The paper contains way too many typos and grammatically incorrect sentences, making it quite distracting to read.

---

> ### Author Rebuttal · Authors · 2025-07-27
>
> We sincerely appreciate your valuable comments on the weaknesses in our manuscript's writing issues. Due to space limitations, we were unable to provide sufficient details in the method and experiment sections. Here, we provide detailed explanations and clarifications in response to your questions Q1–Q4.
> ***
> # Response to Q1
> > Equation (17): Can you explain why you chose the total variation loss rather than an L1 or L2? A brief justification of this design choice would help clarify its role.
> >The Methodology section lacks clarity. The authors simply list a series of equations without explaining the intuition or reasoning behind these design choices, making the section very difficult to follow.
>
> **Reason for loss choice**
> Different from static scenes, objects in high-speed moving scenes are constantly changing, thus the segment of spikes in a time interval is a continuously changing sequence. L1 or L2 loss treats each frame separately, without considering how things move over time. In contrast, TV loss helps keep motion smooth and consistent over time. TV loss looks at the differences between frames. It measures how much the values change on average, and how much they vary. This helps make the output more stable and smooth, but still allows sharp motion when needed.
>
> **Comparison Experiments between losses**
> To further validate our motivation, we conducted comparative experiments on the synthetic dataset, evaluating the average PSNR and SSIM under L1, L2, and temporal TV losses, as shown in Table.R1.
>
> #### **Table.R1 Impact of Spike Loss Function Choice in SPSS.**
> |Metric|L1|L2|Our TV Loss|
> |:----:|:--:|:--:|:--------:|
> |Avg PSNR↑|26.87|26.75|**27.74**|
> |Avg SSIM↑|0.906|0.905|**0.912**|
> ***
> # Response to Q2
> >MSDI is very interesting to me. It would be helpful if the authors could provide a more detailed evaluation of this module—for example, by feeding the MSDI outputs (point clouds, reconstructed images, and poses) directly into 4DGS without SPSS. This would help identify whether the gains come primarily from better pointclouds/poses (versus DUSt3R) or better images (versus spk2img).
>
> Thank you for your attention to MSDI! Regarding your second question, we have actually reported the effect of MSDI combined with 4DGS without SPSS in Table 3. The results demonstrate that, as an initialization module, MSDI outperforms TFI+colmap and DUSt3R in our Spike4DGS pipeline, and the SPSS module further improves the performance of each pipeline.
>
> In light of your insightful comment, **we conducted additional experiments to investigate this aspect**. Specifically, we feed the MSDI outputs (i.e., reconstructed images, point clouds and poses) into standard 4DGS procedure respectively to validate the contribution of MSDI.  In addition,  the traditional spike-to-image methods (e.g., TFI) and MSDI+SPSS are also listed for comparisons. The experiments are conducted on the synthetic dataset and the average performances are reported in the following Table.R2. It indicates both the image and point clouds/ camera poses of MSDI contribute positively.
>
> #### **Table.R3 Comparison of dynamic reconstruction methods with different image sources(PSNR/SSIM) without SPSS**
> | Method |Avg PSNR↑  |Avg SSIM↑ |
> |---------------------------------|-------|-------|
> | TFI + COLMAP + 4DGS             | 21.08 | 0.849 |
> | TFI + DUST3R + 4DGS             | 25.66 | 0.893 |
> | MSDI IMAGE + COLMAP + 4DGS      | 22.43 | 0.854  |
> | MSDI IMAGE + DUST3R + 4DGS      | 25.76 | 0.898|
> | MSDI (IMAGE + POINTCLOUD/POSE) + 4DGS       | **26.66** | **0.903** |
> | MSDI (IMAGE + POINTCLOUD/POSE) + SPSS (Ours) | **27.74** | **0.912**|
> ***
> # Response to Q3
> >Can you also try replacing only the initialization pointcloud with one from DUSt3R[4] or COLMAP[5] while keeping everything else the same? This would help isolate the contribution of your initialization strategy, as any Gaussian Splatting-based rendering method is greatly reliant on the initial point cloud input.
>
> We fully agree that this is an important experiment to assess the impact of the initialization point cloud quality on the final reconstruction results. To this end, **we conducted additional experiments where we replaced only the initialization point cloud and camera poses**.
> Based on this setting, we conduct the experiments on the synthetic dataset, and the average performances are reported in the following Table.R4.
>
> #### **Table.R4 Our full Spike4DGS on different initialization point clouds.**
> |Methods|Avg PSNR↑|Avg SSIM↑|
> |--------|-------|----|
> |MSDI IMAGE + COLMAP+SPSS|23.68|0.867|
> |MSDI IMAGE + Dust3R+SPSS|26.92|0.907|
> |MSDI +SPSS (Full Model)|**27.74**|**0.912**|
> ***
> # Response to Q4
> >Could you show case some examples in more realistic settings—such as a car driving down a road in real lighting conditions? I understand that your pipeline assumes uniform lighting, but seeing examples in real-world scenarios would help reviewers better assess its effectiveness and generalizability. I may reconsider my final rating depending on these results.
>
> We thank the reviewer for the helpful suggestion. **In response, we collected a new scene dataset in an outdoor environment under natural lighting.** We used the dataset to reconstruct some motion examples in real daylight. This new dataset is similar to the scenes in the synthesis data, such as a moving car and a running pedestrian. But the lighting is uneven, and the background is not controlled. These examples add to our previous real-data results. They show that our method can handle complex scenes with changing light and outdoor noise.
>
> Due to rebuttal policies, we are unable to include images. Instead, we provide a table summarizing avg BRISQUE and NIQE on those realistic scenes collected outdoors under natural lighting conditions in Table.R5.
>
> #### **Table.R5 Quantitative Results (BRISQUE↓ / NIQE↓) on More Real Spike-Captured Dynamic Scenes.**
> | Method             | Moving Car  | Running Pedestrian | Moving Bicycle  |
> |--------------------|----------------------------------------|-----------------------------------------------|--------------------------------------------|
> | TFI + D3DGS  | 60.12 / 17.32                  | 58.78 / 15.98                         | 59.27 / 15.04                       |
> | TFI + STG    | 43.26 / 13.20                  | 46.86 / 13.76                        | 43.52 / 13.15                       |
> | TFI + 4DGS   | 35.12 / 11.24                 | 35.37 / 12.05                         | 33.86 / 12.14                       |
> | Spike4DGS          | **30.63 / 10.32**                  | **31.12 / 9.89**                          |**29.95 / 10.01**                       |
>
> These results demonstrate that our method remains robust even without assuming ideal lighting conditions. In future work, we will continue to explore reconstruction performance under more realistic scenarios.
>
> ***
> ##### [1] Lin Zhu, Siwei Dong, Tiejun Huang, and Yonghong Tian. A retina-inspired sampling method for visual texture reconstruction. In 2019 IEEE International Conference on Multimedia and Expo (ICME), pages 1432–1437. IEEE, 2019.
> ##### [2] Lin Zhu, Siwei Dong, Jianing Li, Tiejun Huang, and Yonghong Tian. Retina-like visual image reconstruction via spiking neural model. In IEEE Conference on Computer Vision and Pattern Recognition (CVPR), pages 1438–1446, 2020.
> ##### [3] Jing Zhao, Ruiqin Xiong, Hangfan Liu, Jian Zhang, and Tiejun Huang. Spk2imgnet: Learning to reconstruct dynamic scene from continuous spike stream. In Proceedings of the IEEE/CVF Conference on Computer Vision and Pattern Recognition, pages 11996–12005, 2021.
> ##### [4] Shuzhe Wang, Vincent Leroy, Yohann Cabon, Boris Chidlovskii, and Jerome Revaud. Dust3r: Geometric 3d vision made easy. In CVPR, 2024.
> ##### [5] Johannes Lutz Schönberger and Jan-Michael Frahm. Structure-from-motion revisited. In Conference on Computer Vision and Pattern Recognition (CVPR), 2016.
> ##### [6] Guanjun Wu, Taoran Yi, Jiemin Fang, Lingxi Xie, Xiaopeng Zhang, Wei Wei, Wenyu Liu, Qi Tian, and Xinggang Wang. 4d gaussian splatting for real-time dynamic scene rendering. In Proceedings of the IEEE/CVF Conference on Computer Vision and Pattern Recognition, pages 20310–20320, 2024.

---

> > ### Comment · Area_Chair_xHAy · 2025-08-08
> >
> > Dear Reviewer,
> >
> > Thank you for the thoughtful initial reviews.
> > The author has submitted the rebuttal. Please review it promptly and, before making your final decision, make every effort to resolve any questions or concerns through discussion with the author.
> >
> > Best regards,
> >
> > Your AC

---

### Official Review · Reviewer_KHNA · 2025-07-08

**Clarity:** 3
**Significance:** 2
**Originality:** 3
**Rating:** 4
**Confidence:** 3

**Summary:**

This paper proposes Spike4DGS, the first high-speed dynamic scene rendering framework with 4D Gaussian Splatting using spike camera arrays. 3DGS from conventional cameras is fundamentally limited by framerate, and therefore struggle with fast-moving scenes. Spike cameras asynchronously encode absolute light intensity into continuous spike streams at rates of up to 20k Hz. However, prior works with spike cameras are limited to 3DGS: thereby suffering in dynamic scenes.

To solve this, the input multi-view spike streams are sent to a Multi-view Spike-based Dense Initialization to estimate point cloud and camera poses. MSDI consists of ViT based encoders which are fused to generate a point cloud initialization and pose estimates. The features are also used to generate discrete images from the spike stream. Using these as inputs to a 4DGS generation pipeline, a 4DGS model may be generated. To better supervise 4DGS training, in addition to the pixel level loss, rendered images are also converted to spike streams, which can be supervised against the original spike stream.

The authors propose both a synthetic and real multi-view dataset of spike streams. On this dataset, the proposed method outperforms comparison methods in terms of novel view rendering quality on both synthetic and real-world datasets. Ablation studies show the benefit of various components in the method.

**Questions:**

1. The text in the figure 4 results seems flipped. Why is this?
2. Can we have some sense of training time, and how it compares with baseline methods?
3. Is there any reason why all the cameras were arranged in a planar manner?

**Ethical Concerns:**

["NO or VERY MINOR ethics concerns only"]

**Final Justification:**

I would like to thank the authors for the response and the discussion. After reading the comments from other reviewers, and the authors responses, I would like to weight towards acceptance and lean towards a 'weak accept' score, in context of the strengths and weaknesses of this work.

**Limitations:**

Yes, with some suggestions in the answers above

**Quality:**

3

**Strengths And Weaknesses:**

Strengths:
1. The paper is written well, and it is reasonably straightforward to understand the method and contributions
2. The motivation for the method, in terms of capturing high-dynamic scenes using spike cameras, is well described
3. This paper proposes the first 4DGS method for spike cameras
4. Results show superior performance against compared baselines
5. Ablations motivate the need for various method components well

Weaknesses:
1. Since this is a 4DGS method, I think it is imperative to see several novel views for a given scene to show that the generated 3D representation is indeed robust, which I could not find in the paper.
2. Overall, the method seems largely sequential. First, spike camera views are converted into traditional RGB camera representations (images, point clouds, camera poses etc.), and these are passed through an existing 4DGS method. So it seems the major novelty is the conversion of spike camera views to RGB views.
3. The training camera views are all planar and therefore cannot truly represent the entire 3D scene, meaning the 3D representation quality on faraway camera views cannot be tested.
4. It is not clear what type of scenes this method is good at representing vs what types of scenes it is not good for: the limitations section should incorporate this.

---

> ### Author Rebuttal · Authors · 2025-07-27
>
> Thank you for your positive and thoughtful comments. We are delighted that you find our idea very important and the results great success and impressive. We would like to address your concerns and answer your questions in the following.
>
> # Respond to W1
> >Since this is a 4DGS method, I think it is imperative to see several novel views for a given scene to show that the generated 3D representation is indeed robust, which I could not find in the paper.
>
> In fact, for both the real-world and synthetic datasets, the images shown in our Figures 3, 4, and 5 comparisons are rendered from a novel view. Specifically, for the real-world dataset, we employ a 3×3 spike camera array. For the synthetic dataset, we simulate the same 3×3 camera grid. In both cases, we select the central view as the novel test view and other 8 views as training views. As shown in Figures 3, 4, and 5,  the novel view performances demonstrate that our method maintains reasonable robustness, consistent with the goals of a 4DGS framework.
>
> # Respond to W2
> >Overall, the method seems largely sequential. First, spike camera views are converted into traditional RGB camera representations (images, point clouds, camera poses, etc.), and these are passed through an existing 4DGS method. So it seems the major novelty is the conversion of spike camera views to RGB views.
>
> We appreciate the reviewer’s observation. While our pipeline does leverage the strength of existing 4DGS techniques for scene representation and rendering, we emphasize that our contribution goes far beyond a simple conversion from spike views to RGB representations.
>
> **Introduction of spike variation supervision**
> Apart from MSDI, we also propose a Spike-Pixel Synergy Supervision, which introduces a spike variation supervision.
> This loss encourages local consistency in the temporal dimension, effectively reducing noise in the spike stream without excessively blurring the true motion.  It measures how much the values change on average, and how much they vary. This helps make the output more stable and smooth, but still allows sharp motion when needed.
> To further validate its effectiveness, we conducted comparative experiments on the synthetic dataset, evaluating the average PSNR and SSIM. We compare our spike total variation loss (TV loss) with naive L1 loss in spikes, as shown in Table.R1. Our spike TV loss in SPSS gets the SOTA result.
>
> #### **Table.R1 Impact of Spike Loss Function Choice in SPSS.**
> ||Avg PSNR ↑ |Avg SSIM ↑ |
> |---------------------|--------|--------|
> | Ours(TV loss)| **27.74**| **0.912**  |
> | w/o TV loss,use L1 loss|26.87| 0.906  |
> | w/o any spike loss | 26.66  | 0.903 |
>
> **Introduction of MSDI**
> We introduce MSDI, a framework tailored for spike-based 3D vision, which is fundamentally different from conventional RGB pipelines such as Dust3R. MSDI includes not only a module for converting spike data into images, but also designs for camera pose estimation and 3D geometry reconstruction directly from spike modality. We quantitatively evaluate the camera pose estimation accuracy of MSDI in comparison with Colmap and Dust3R [1] on the synthesis dataset. The ground truth camera poses could be obtained from the Carla simulator. The Rotation Error (°)  and Translation Error are computed as same as Dust3R [1]. The results are shown in the table below Table.R2:
>
> #### **Table.R2 Comparison of Camera Pose Estimation results.**
> | Method           | Rotation Error (°) ↓ | Translation Error (%) ↓ | Time  ↓ |
> |------------------|----------------------|--------------------------|------------|
> | TFI+COLMAP | 2.72                 | 9.8                      | 30.3s       |
> | TFI+Dust3R   | 1.27                 | 3.5                      | 9.6s       |
> | Ours (MSDI)      | **0.94**             | **2.1**                  | **9.1s**    |
>
> # Respond to Q1
> > The text in Figure 4 results seems flipped. Why is this?
>
> The observed image flip arises from differences in the data acquisition devices. The raw spike data is collected using a spike camera array, whereas the last column of RGB images is captured by a handheld RGB mobile phone, solely for the purpose of illustrating the real-world scenes. These RGB images are not used in model training or in the rendering pipeline.
>
> The flip is not caused by our model or the rendering process. Instead, it results from a transpose operation applied during the preprocessing of spike data. In contrast, the RGB images are captured directly without any such operation. This discrepancy has no impact on the reconstruction results.
>
> # Respond to Q2
> > Can we have some sense of training time, and how it compares with baseline methods?
>
> For this question, we compiled a comparison table of the training time of our method and the baseline methods on our synthesis dataset. We report the training time and the average PSNR of each method on one Nvidia RTX 4090, as shown in Table R.3.
>
> #### **Table.R3 Training Time and the Average PSNR of Each Method**
> | Methods  | TFI+STG[2]  | TFI+4DGS [3] | TFI+D3DGS[4]  | Ours |
> |----------|-------|-------|-------|-----------|
> | Avg time | 24min | 22min | 19min | 25min     |
> | Avg PSNR     | 24.70   | 26.00   | 20.34 |  **27.74** |
>
> While our method achieves similar training efficiency, its key contribution lies in the MSDI module and dual-modality supervision design. Both of them significantly improve the reconstruction results.
> # Respond to Q3
> > Is there any reason why all the cameras were arranged in a planar manner?
>
> In practice, we used a 3×3 camera array and ensured that the object was centered directly in front of the middle camera. All surrounding cameras were tilted inward and oriented toward the center camera, forming a conical viewing region with significant overlap among the fields of view. This configuration ensures that each scene is well covered from multiple angles, which is reasonable for successful 3D reconstruction. The dataset will be released with the publication of the paper.
>
> ***
> ##### [1] Shuzhe Wang, Vincent Leroy, Yohann Cabon, Boris Chidlovskii, and Jerome Revaud. Dust3r: Geometric 3d vision made easy. In CVPR, 2024.
> ##### [2] Zhan Li, Zhang Chen, Zhong Li, and Yi Xu. Spacetime gaussian feature splatting for real-time dynamic view synthesis. In Proceedings of the IEEE/CVF Conference on Computer Vision and Pattern Recognition, pages 8508–8520, 2024.
>
> ##### [3] Guanjun Wu, Taoran Yi, Jiemin Fang, Lingxi Xie, Xiaopeng Zhang, Wei Wei, Wenyu Liu, Qi Tian, and Xinggang Wang. 4d gaussian splatting for real-time dynamic scene rendering. In Proceedings of the IEEE/CVF Conference on Computer Vision and Pattern Recognition, pages 20310–20320, 2024.
>
> ##### [4] Jonathon Luiten, Georgios Kopanas, Bastian Leibe, and Deva Ramanan. Dynamic 3d gaussians: Tracking by persistent dynamic view synthesis. In 2024 International Conference on 3D Vision (3DV), pages 800–809. IEEE, 2024.

---

> > ### Comment · Area_Chair_xHAy · 2025-08-08
> >
> > Dear Reviewer,
> >
> > Thank you for the thoughtful initial reviews.
> > The author has submitted the rebuttal. Please review it promptly and, before making your final decision, make every effort to resolve any questions or concerns through discussion with the author.
> >
> > Best regards,
> >
> > Your AC

---

> ### Comment · Reviewer_KHNA · 2025-08-09
> **Response**
>
> I would like to thank the reviewers for their thorough response. My questions have been largely addressed through this response, and my recommendation will weigh towards acceptance. I will assign my final score after discussion with other reviewers.

---

### Note · Authors · 2025-08-12

Thank all reviewers and the AC for their time and effort. We sincerely appreciate some thoughtful comments and suggestions provided during the review and discussion process.

(1) Regarding the dataset, due to the rebuttal policy, we could not provide an access link during the rebuttal phase. We confirm that we will release both the dataset and the code after publication. All datasets are strictly for academic use, and the data collection process involved no ethical concerns.

(2) We highly value the insightful feedback from each reviewer, and we will ensure that all our responses are included as supplementary material in the revised version.

(3) Regarding reviewer xVW6, we sincerely hope our rebuttal has addressed his/her concerns. If there are any remaining questions, we would be happy to provide further clarification.

---

### Decision · Program_Chairs · 2025-09-17

**Decision:**

Accept (poster)

**Comment:**

This paper presents Spike4DGS, a framework based on 4DGS for reconstructing dynamic scenes using a spike camera array. The method employs MSDI (Multi-view Spike-based Dense Initialization) to jointly estimate images, point clouds, and poses, and introduces SPSS (Spike-Pixel Synergy Supervision) to leverage the high temporal resolution of spike signals in the training loss. The authors also construct both synthetic and real-world datasets, which they plan to release.

In the rebuttal, the authors provided detailed validations of MSDI and SPSS, additional evaluations on more practical real-world scenes under natural illumination, quantitative evaluations of geometric reconstruction accuracy, and comparisons with further baselines. These clarifications and experiments largely addressed the reviewers’ concerns, and all reviewers converged on positive ratings.

Overall, this paper demonstrates the effective application of spike camera sensing to dynamic 4DGS. The planned dataset release will likely serve as a valuable resource for future research. I recommend acceptance.

The AC encourages the authors to include the additional experiments provided in the rebuttal, including the visual analyses, in the camera-ready version.